# Dynamic network coding of working-memory domains and working-memory processes

Eyal Soreq [1], Robert Leech [2] & Adam Hampshire[1]

The classic mapping of distinct aspects of working memory (WM) to mutually exclusive brain areas is at odds with the distributed processing mechanisms proposed by contemporary network science theory. Here, we use machine-learning to determine how aspects of WM are dynamically coded in the human brain. Using cross-validation across independent fMRI studies, we demonstrate that stimulus domains (spatial, number and fractal) and WM processes (encode, maintain, probe) are classifiable with high accuracy from the patterns of network activity and connectivity that they evoke. This is the case even when focusing on 'multiple demands' brain regions, which are active across all WM conditions. Contrary to early neuropsychological perspectives, these aspects of WM do not map exclusively to brain areas or processing streams; however, the mappings from that literature form salient features within the corresponding multivariate connectivity patterns. Furthermore, connectivity patterns provide the most precise basis for classification and become fine-tuned as maintenance load increases. These results accord with a network-coding mechanism, where the same brain regions support diverse WM demands by adopting different connectivity states.

[1] The Computational, Cognitive and Clinical Neuroimaging Laboratory, Division of Brain Sciences, Imperial College London, London W12 0NN, UK. [2] Centre for Neuroimaging Sciences, Institute of Psychiatry, Kings College London, London SE5 8AF, UK. Correspondence and requests for materials should be addressed to E.S. (email: e.soreq14@imperial.ac.uk)

I t is well-established that human working memory (WM) is behaviourally complex, comprising dissociable systems for temporarily maintaining different types of information in mind[1,2] and mechanisms for selectively attending and transforming that information[3,4]. A major aim of early functional imaging research was to map these distinct aspects of WM onto the brain. Initially, a localist framework was applied, which sought to understand WM in terms of discrete brain circuits, where each neuroanatomical component was ascribed a specific function[5,6]. WM stages, including encoding[7], maintenance[8] and probe[9], WM demands, including maintenance load[10–13] and mental manipulation[7,14], and WM domains, including spatial, object or numeric[5,15,16], were attributed to dedicated brain regions and processing streams[17].

More recently, advances in connectivity methods have motivated a paradigm shift towards a network-coding perspective, where cognitive processes are considered emergent properties of interactions that occur across different widespread coalitions of brain regions[18–24]. From this perspective, the localist framework is too simplistic because WM functions map to underlying brain systems in a many-to-many as opposed to one-to-one manner[25,26]. This notion of network coding has gained rapid support[26,27]; however, it remains unclear how aspects of WM that are known to be behaviourally distinct[1–4] map within the multivariate network space. Furthermore, there is a need to reconcile the network perspective with the robust evidence of mutually exclusive localist mappings of WM functions, as provided by the early-brain imagining literature[5–17].

We addressed these issues by developing a multivariate machine-learning pipeline to examine how patterns of brain activity and connectivity dynamically changed during the performance of two novel fMRI tasks that were designed to probe distinct aspects of WM function[9,28–31]. We tested key predictions of the network perspective while seeking to understand how evidence of classic localist mappings can coexist with it. To avoid bias all analyses were data driven and to ensure reproducibility the results were robustly confirmed across datasets from independent studies. We predicted that: (1) No brain region or connection would be involved exclusively in one or other aspect of WM. (2) WM stages (encode, maintain, or probe) and domains (spatial, number or fractal) would be reliably classifiable from the multivariate patterns of brain activity and connectivity that they evoked. (3) Classification accuracies for these aspects of WM would be significant even when focusing on 'Multiple Demand Cortex', i.e., the set of brain regions that is most consistently active across cognitive conditions[32,33]. (4) Classic localist mappings such as dorsolateral and left lateralised frontal cortex involvement in spatial and number WM would be evident as non-exclusive features within these multivariate patterns. (e) Conditions that affect WM difficulty, e.g., maintenance load, would relate most closely to network dynamics[11–13] as opposed to regional brain activity[34,35]. Our results strongly support the network-coding perspective, provide new insights into the underlying mechanisms of WM and demonstrate how models of the early localist literature may be reconciled.

## Results

**Task design and behavioural results**. The behavioural task designs are detailed in Fig. 1. In brief, during Study 1, participants were presented with arrays containing numbers and fractals displayed at a subset of spatial locations within a $4 \times 4$ grid. They had 10 s to encode the items from one of these three-dimensions and then maintained that information for a further 10 s delay. Subsequently, a probe array was displayed that differed with respect to one number, one fractal-pattern and one location.

Participants identified the non-matching item from the maintained domain. A 10 s inter-stimulus-interval (ITI) separated the next trials. The number of items from each dimension (WM load) was 3, 5 or 7.

Accuracies (Supplementary Tables 1, 2 and Supplementary Figure 1) were examined in a $3 \times 3$ (load × domain) repeated-measures ANOVA. Mean accuracy was 91.13% ± 6.31 SDs. There was a significant main effect of load ($F_{(2,36)} = 11.85$ $p < 0.001$) with lower accuracy at high load (high < low $t = 4.859$, $p < 0.001$; high < medium $t = -3.03$, $p < 0.018$). The main effect of domain and load × domain interaction was non-significant. Reaction times (RT) (Supplementary Tables 3, 4 and Supplementary Figure 2) were examined in a model of the same structure. Median reaction times (RT) (correct trials only) were 3.637 s ± 1.54 SDs. There were significant main effects of load and domain (load $F_{(2,36)} = 168.59$, $p < 0.0001$; domain $F_{(2,36)} = 24.5$, $p < 0.0001$), and a significant load × domain interaction ($F_{(4,72)} = 4.508$, $p < 0.005$). RTs increased with load ($t$ low < med = $-10.7$, $p < 0.0001$; $t$ low < high = $-16.3$, $p < 0.0001$). Fractal and number trials were slower than location trials ($t$ location < number = $-5.38$, $p < 0.001$; $t$ location < fractal = $-5.88$, $p < 0.001$).

Study 2 used the same task design, except that (1) only numbers were displayed on the grid (2) there were two levels of WM load (3 & 6) and (3) a retro-cue indicating either 'maintain' or 'manipulate' preceded the maintenance period. In the manipulate condition, participants transformed the information by spatially rotating the positions 90 degrees clockwise or incrementing each number by 3. The retro cue was pseudo-randomised to ensure that encoding demands matched across the maintain and manipulate trials.

Accuracies (Supplementary Tables 5, 6 and Supplementary Figure 3) were examined in a $2 \times 2 \times 2$ (manipulation × domain × load) repeated-measures ANOVA. Mean accuracy was 84.24% ± 8.5 SDs. There was a significant main effect of load ($F_{(1,15)} = 43.2$, $p < 0.0001$) and a significant load × domain interaction ($F_{(1,15)} = 5.29$, $p < 0.04$) with accuracy for number trials lower during high-load. RTs (Supplementary Tables 7, 8 and Supplementary Figure 4) were examined in a model of the same structure. Median reaction times (RT) (correct trials only) were 3.937 s ± 1.26 SDs. There was a significant main effect of load and a significant load × domain interaction ($F_{(1,15)} = 74.26$, $p < 0.0001$; $F_{(1,15)} = 17.05$, $p < 0.001$). RTs increased as a function of load ($t = 8.60$, $p < 0.0001$), and RTs for number and location trials under low load were slower relative to high load ($t = 9.17$, $p < 0.001$; $t = 4.7$, $p < 0.001$). There were no significant effects of manipulation and no other significant interactions. Together, these results show that participants were able to perform both tasks with high accuracy and that there were the expected costs of WM load on performance.

**Mapping multiple demand brain regions**. Single subject general linear models capturing voxel-wise changes in activity were constructed using a mini-block design (Fig. 1e), in which each stage (e.g., cue, encode, maintain, probe) of each trial was captured by a separate predictor, alongside nuisance covariates. Conjunction analyses[36] (i.e. logical AND, Fig. 2a, d) were conducted to identify regions that were consistently active across different WM conditions (i.e. stimuli domains and processing stages). To identity brain areas that were jointly active for all domains (Domain General—DG) we performed a conjunction between the number, fractal and location trials using parameter estimates averaged across WM stages and loads. The resulting activation pattern resembled the brain volume commonly referred to as 'Multiple Demand Cortex'[32,33] (Fig. 2b). To identify the subset of DG regions where activity was sustained during all three stages (e.g., encode, maintain and probe) a second conjunction

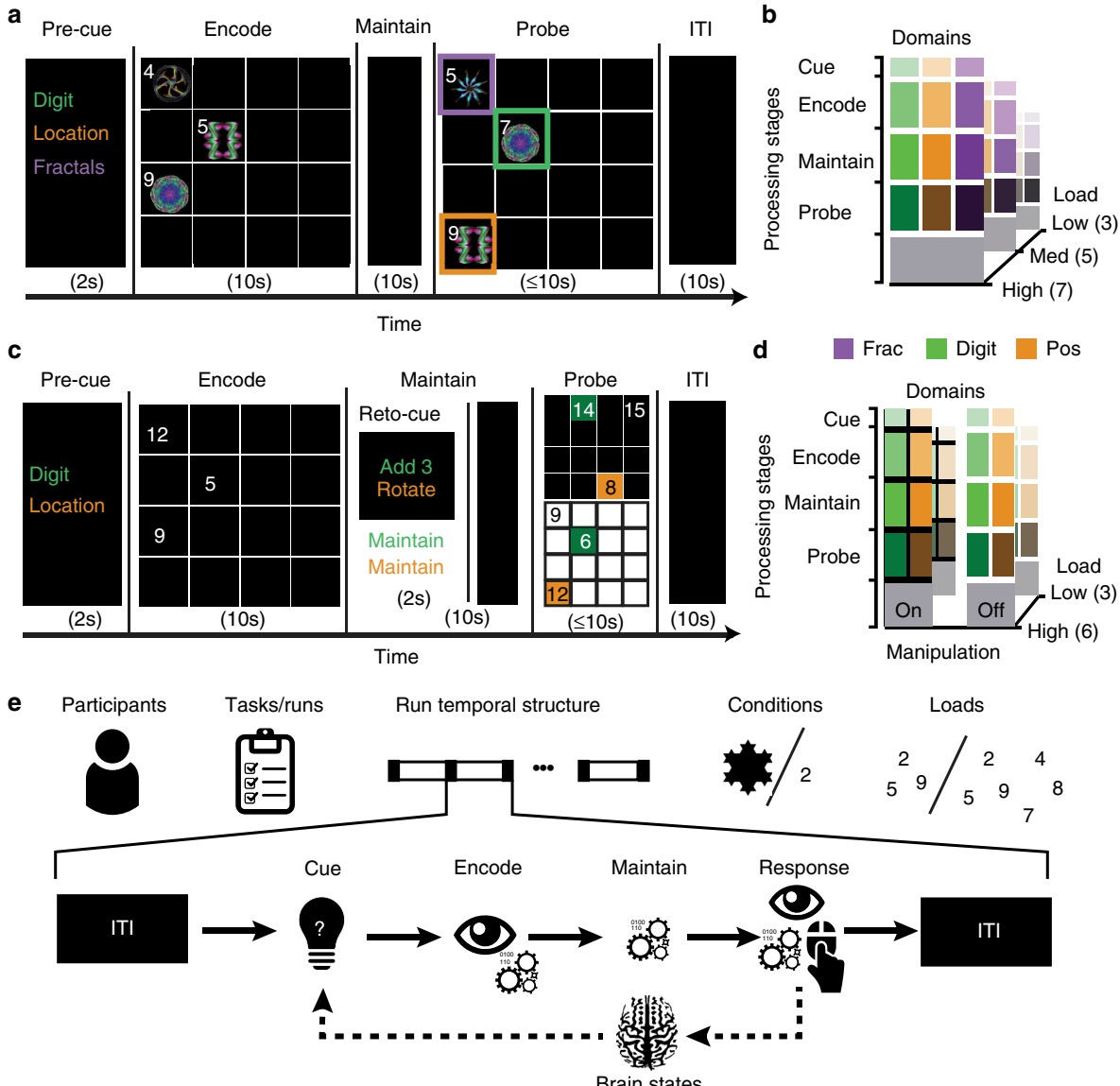

**Fig. 1** Study and task design. Participants undertook three runs of the task consisting of contiguous sequences of WM trials presented in pseudo-randomised order. **a** Each trial in Study 1 had four processing stages. (1) A pre-cue informed the participant to encode either numbers, fractals or spatial locations. (2) An array was displayed containing numbers and fractal patterns co-located within a 4 × 4 grid. The participant had 10 s to encode the stimuli from the cued domain. (3) The stimuli were replaced with a fixation cross that was displayed for 10 s. The participant maintained the encoded information in WM during this delay. (4) The stimulus array reappeared with one item from each domain (number, fractal and location) changed. The participant tried to identify the cell that contained the changed item from the maintained domain. **b** The trials differed according to cued stimulus domain and number of items (3, 5 or 7) per domain (load) in a 3 × 3 factorial design. **c** Study 2 differed in three ways. (1) There were two stimulus domains—numbers and spatial locations. (2) There were two levels of WM load (3 vs. 6). (3) A retro-cue was displayed at the start of the maintenance period. On 50% of trials this informed the participants to maintain the encoded information. On the other 50% it instructed them to manipulate that information, i.e., by mentally rotating the encoded spatial positions 90 degrees clockwise or adding 3 to encoded numbers. **d** The trials differed according to cued stimulus domain, number of items per domain and requirement to manipulate in a 2 × 2 × 2 factorial design. **e** The trials were separated by a 10 s inter-trial-interval (ITI). Measures of activity and connectivity were calculated separately for each trial, stage and participant; these data formed the input to the machine-learning pipeline

was performed using parameter estimates averaged across WM domains and loads. We refer to the resultant map as Stage General (SG) (Fig. 2c). This division of DG into SG and non-SG produced a distinct separation between areas that were active for specific processing stages, including primary, secondary, associative visual cortex, motor areas and bilateral thalamus, and areas that were active for all stages, which included, lateral frontal and parietal cortices, anterior insula and thalamus. Finally, we confirmed the reproducibility of the DG and SG conjunctions across the two studies by calculating dice coefficients, which provide a binary estimate of similarity between the patterns of

activation. High correspondences were evident: DG = 0.80 and SG = 0.76.

**Regions of interest analysis.** To reduce computational load voxels within the brain were combined as regions of interest. This data-reduction step is important for network connectivity analyses which produce a high number of features per node[37]. Key predictions pertained to Multiple Demand Cortex; therefore, we used a custom three-dimensional variant on the watershed algorithm[38] to parcellate the SG and DG-conjunction maps from Study 1 into discrete activation clusters comprising 20 and 33

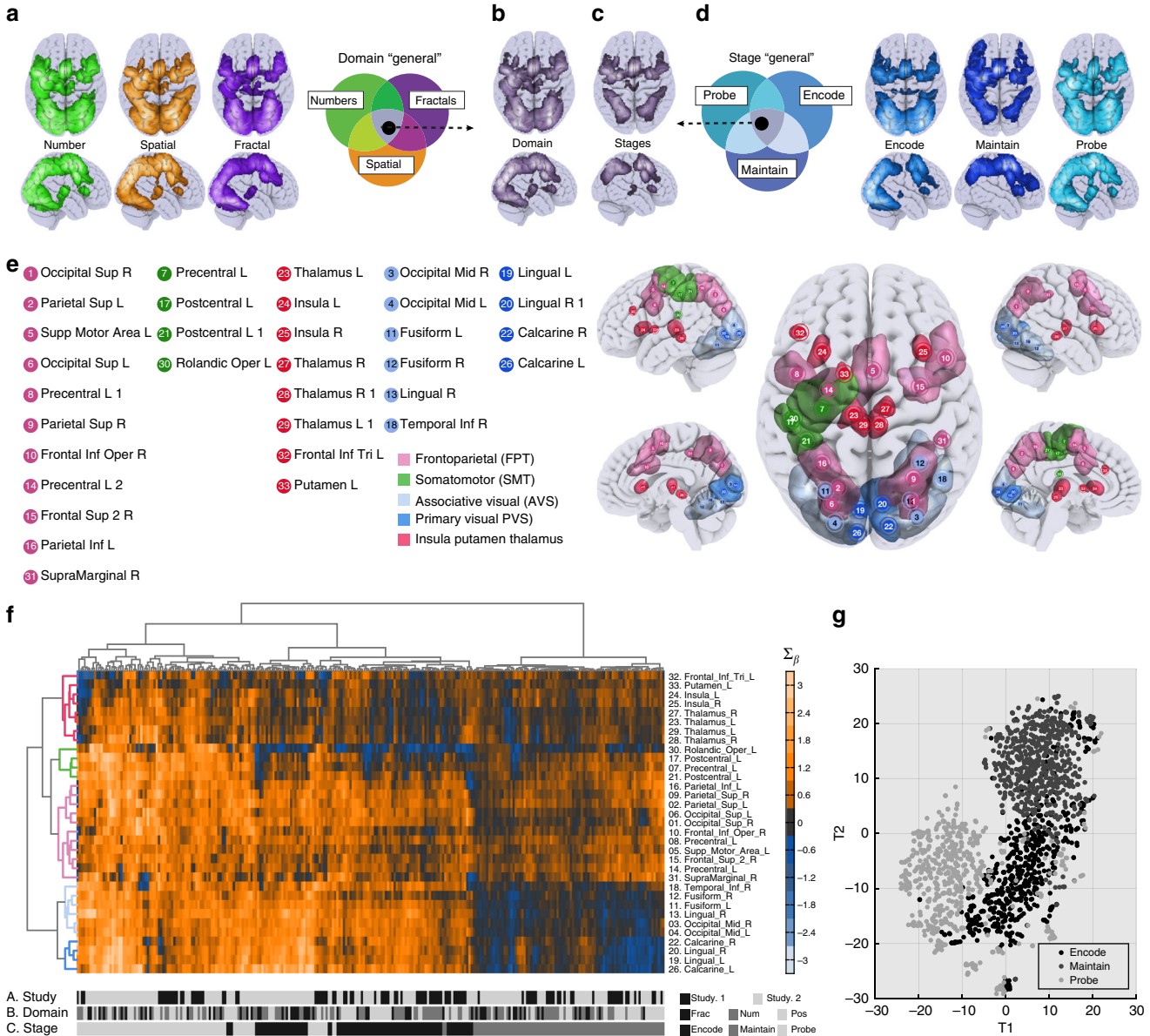

**Fig. 2** Inspecting activation commonalities across experiments. To examine the similarity in voxelwise activations across the two studies, we performed two-independent conjunction analyses focusing on WM processing stages and domains. Analysing the T-statistics maps for the three stimulus domains compared to ITI period (**a**) a minimal statistics intersection produced (**b**) the domain general (DG) map. Conducting a further intersection analysis on T-statistics maps for the processing stages (**d**) produced (**c**) the subset of DG voxels that were 'state general', i.e., being active for all stages of the WM task (SG). **e** Clustering of ROIs (rows) produced an interpretable functional separation consisting of frontoparietal, somatomotor, associative visual, and primary visual areas, and insular plus striatal areas. **f** Hierarchical clustering analysis was applied to brain activation data with WM stages per domain separately on the X-axis and DG ROIs on the Y-axis. Clustering (represented by similar coloured sections of the dendrogram) of these mini-blocks (columns) corresponded to the WM stages not stimulus domains and had high purity (94.63%). **g** Applying dimension reduction to a whole-brain regions of interest further supported that WM stages formed the purest clustering of ROI activation patterns

ROIs respectively (Supplementary Tables 9, 10). For completeness, we also conducted analyses with an established whole-brain (WB) parcellation[39] covering the entire brain with 268 ROIs (Supplementary Table 11).

**Unsupervised clustering based on ROI activity.** Hierarchical clustering[40] was used to simultaneously group in a data-driven manner the DG ROIs according to similarity of their activation profiles across WM conditions, and the WM conditions according to similarity of their associated patterns of activation across the DG ROIs. This analysis demonstrated high dissociability of WM stages (Fig. 2e, Total Purity = 94.01% see methods) relative

to WM domains. There was an interpretable fractionation of the DG ROIs into five distinct clusters corresponding to (1) frontoparietal, (2) somatomotor, (3) associative visual, (4) primary visual and (5) insula, putamen and thalamus (Fig. 2f). Activity was generally lower during maintenance relative to encoding or probe, and this effect was particularly pronounced for the visual clusters[41]. To examine the dominant natural structure of the data more holistically, t-Distributed Stochastic Neighbour Embedding[42] was used to group all mini-blocks in a data driven manner based on the patterns of activation across the WB ROIs. This analysis again demonstrated that the mini-blocks clustered most strongly in both studies according to processing stage (Fig. 2g).

This dominance of WM stage is expected given the task design, because the stages differ according to the visual and motor demands of the task.

**Maintenance evokes a low-activity high-connectivity state.** To further investigate the lower activity observed during maintenance, ROI activations and functional connectivity (FC) estimates from generalised psychophysiological interaction models[28,30] (PPI) were averaged across regions and connections (Fig. 3a) for each ROI cluster during the encoding and maintenance stages of the task. The frontoparietal ROIs were consistently active during both encoding and maintenance relative to the inter-trial interval (ITI); whereas, the visual ROIs were more

active during encoding than the ITI but not during maintenance (Fig. 3b & Supplementary Table 12). In contrast, intra-cluster connectivity (i.e. averaged for all connections within each cluster), as well as inter-cluster connectivity (i.e. averaged for all connections between clusters), were significantly increased for all clusters ($p < 0.001$) during both encoding and maintenance relative to the ITI; this was the case for the frontoparietal and the visual clusters (Fig. 3c & Supplementary Tables 13, 14). Differences in the strength of intra-cluster connectivity between encoding and maintenance were small/negligible in the frontoparietal and the visual areas, although the latter were somewhat variable across studies (Supplementary Table 15). Inter-cluster connectivity was substantially stronger during encoding than maintenance, although it was significantly greater than resting baseline for both

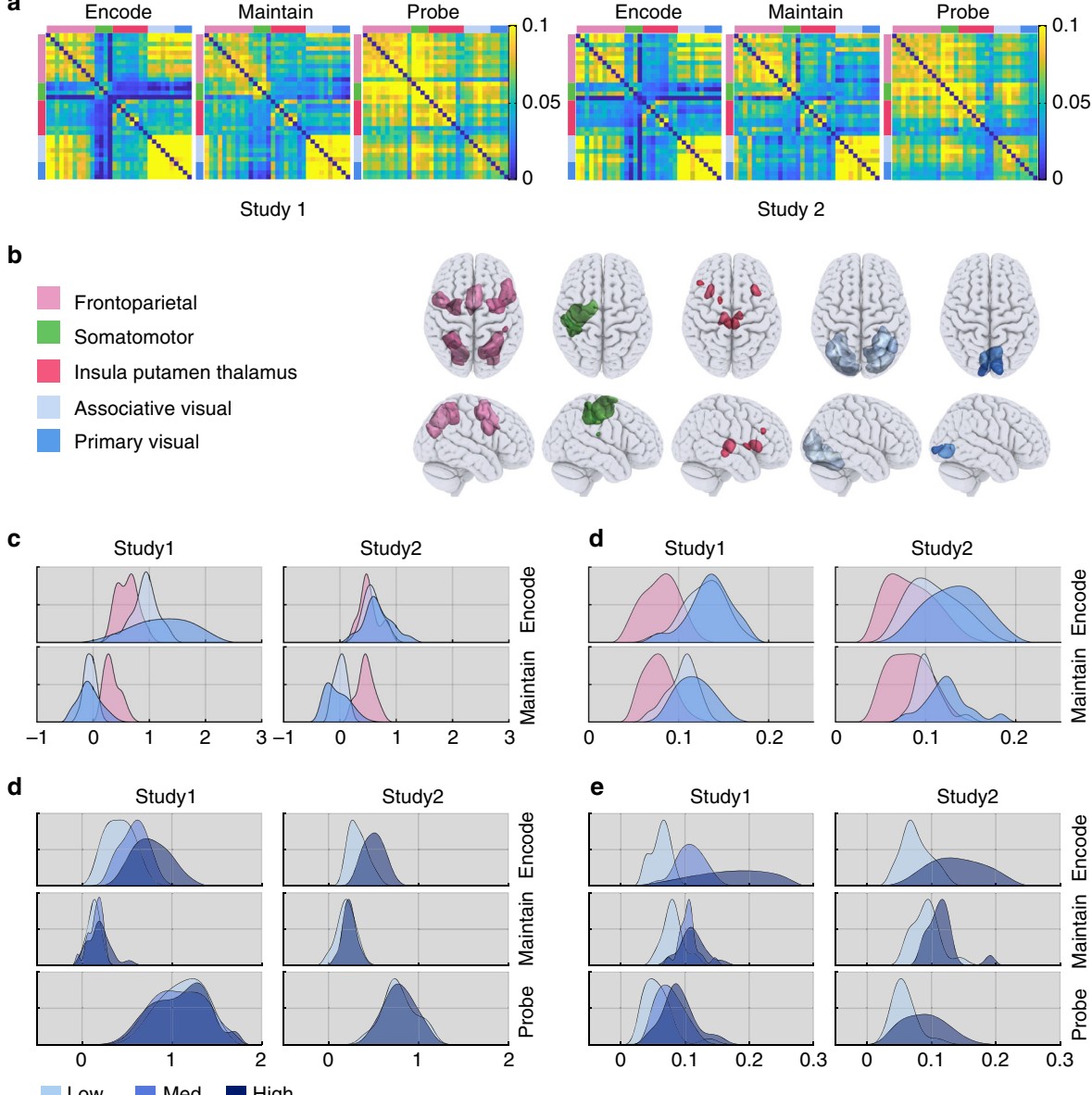

**Fig. 3** Comparing bold activity and functional connectivity across experiments. **a** Mean functional connectivity matrices during encoding, maintenance and probe for the two studies. Connectivity matrices were sorted based on the previously defined functional clusters. Distributions across participants of mean DG ROI activity (**b**) and intra-cluster functional connectivity (**c**) during encoding and maintenance relative to the ITI. Colours represent frontal and visual clusters (FPT, AVS and PVS). All clusters were active during encoding but only the FPT cluster was active during maintenance. All clusters showed significant FC during both encoding and maintenance. Distributions across participants of DG ROI activity (**d**) and intra-cluster functional connectivity (**e**) averaged across ROI cluster and broken down by load. Activity was increased at higher loads during encoding but not during maintenance or probe. In contrast, intra-cluster FC was upregulated at higher WM load during all three stages

(Supplementary Table 16). Thus, WM maintenance is characterised by heightened frontoparietal activity and a stable connectivity state throughout the broader WM network.

**Load-related effects during maintenance**. If the maintenance of WM information related to frontoparietal and visual functional connectivity, then higher maintenance load should have primarily affected inter- and intra-cluster FC during the maintenance period. We tested this by comparing changes in mean activity, as well as intra- and inter-cluster FC during the maintenance stage relative to the ITI. In accordance with our prediction, mean activity across the DG ROIs was insensitive to higher WM load during the maintenance period ($F_{2,3,6} = 0.75$, $p > 0.43$). Analysis of individual ROIs (Bonferroni-Holm corrected) showed load-related increases ($p < 0.0001$) during maintenance amongst a limited subset of brain regions. These included the superior and inferior parietal, precentral, insula and supramarginal cortex ROIs (Supplementary Table 17). In contrast, there was a robust global increase at higher load in intra-FC ($F_{2,3,6} = 22.55$, $p < 0.0001$) (Fig. 3e, Supplementary Tables 18–26). Study 2 showed the same effect ($t_{15} = 2.71$, $p < 0.0161$).

To determine whether a specific subset of connections was upregulated at higher load regardless of domain, a mass univariate analysis was conducted, in which a general linear model comprising domain and load main effects, and their interaction was fitted to each connection. When analysing entire trials (Fig. 4a), significant domain main effects (Fig. 4b) were evident focused on the intra-connections of the frontoparietal ROI cluster along with connections to associative visual and motor ROIs. Load effects were primarily evident for the connections between visual ROIs and their connections to the frontoparietal and insula ROIs. Connections showing a significant interaction between load and domain were more evenly distributed throughout the network. Critically though, when examining data from the maintenance (Fig. 4c) stage, i.e., when there were no differences in the visual stimuli on screen, no main effects of load were evident; instead, there were main effects of domain, and domain by load interactions. These were focused on connections within the frontoparietal ROIs and intra-connections of frontoparietal and visual ROIs. These results accord with the expected prominent role for frontoparietal connections in WM maintenance. They also demonstrate that increased maintenance load leads to selective upregulation of different combinations of connections, i.e. dependent on the type of information that is being maintained.

**Decoding processing stages**. As WM stages showed strong separation in the hierarchical cluster analysis, we expected that multivariate pattern analysis would categorise WM stages with high-accuracy based on either activity or connectivity. A three-way classifier was trained on data from Study 1. The accuracy of the classifier model was determined by applying it to a cross-validation holdout subset from Study 1 and data from Study 2, to which it was naïve. This process was repeated for different ROI-sets (i.e. a whole-brain parcellation of another group (WB)[39], or focused on the DG and SG ROIs). Classification of processing stages based on multivariate activation ($Acc_{WB} = 90.89$; $Acc_{DG} = 91.37$; $Acc_{SG} = 75.89$) (Fig. 4e) and connectivity ($Acc_{WB} = 81.18$; $Acc_{DG} = 87.58$; $Acc_{SG} = 76.13$) (Fig. 4f) patterns was of extreme high accuracy compared to the null models. This was the case for all ROI sets. Comparing models for different ROI-sets showed that the SG ROIs provided a less accurate classification than either the WB or DG sets ($p < 0.001$). This suggests, that while the connectivity within the SG-volume is sufficient to decode stages accurately, there is significant additional information within other

brain areas (e.g., visual and motor) of the DG volume (Supplementary Tables 27–30).

**Decoding visual domains**. The hierarchical clustering showed only weak dissociations between the WM domains; however, the mass univariate analysis showed domain sensitivity of selected connections, which could indicate multivariate coding of domains. We tested whether the domains could be decoded based on the activity or connectivity patterns from any of the ROI sets (WB, DG and SG) during maintenance, i.e., where visual and motor demands were carefully controlled in the task design. The same process was applied as per analysis of the processing stages, with classifiers trained on data from Study 1, then applied to holdout data from Study 1 and data from Study 2, to which they were naive. Remarkably, the classification accuracies for WM domains were comparable to those for the WM stages for ROI activation patterns ($Acc_{WB} = 91.75$; $F1_{DG} = 70.6$; $F1_{SG} = 75.2$) (Fig. 4e) and connectivity patterns ($Acc_{WB} = 87.9$; $F1_{DG} = 85.8$; $F1_{SG} = 83.6$) (Fig. 4f). Post-hoc comparisons showed that added information contained in the WB ROI-set significantly ($p < 0.001$) improved classification compared to both DG and SG sets (Supplementary Tables 31, 32).

**Increased accuracy of decoding at higher WM load**. A key prediction of the network perspective was that the connectivity state of the WM network would become more synchronous at higher load. To test whether this led to more discriminable domains, the classifiers were applied to data stacks comprised of single trial connectivity and activity measures, and accuracy was compared across levels of load during the maintenance stage. The results showed that even when applied to the single trial data, regardless of ROI set or metric, the visual domains could be classified with significant accuracy relative to the permutation null distributions (mean accuracy = 68.2%). More importantly, classification was substantially more accurate for both medium and high load compared to low load, regardless of measure (activity or connectivity) and for both studies. Therefore, as maintenance load increased, domain-related activation (Fig. 4g), and to a greater degree connectivity patterns (Fig. 4h), became more discriminable (both $p < 0.001$, Supplementary Tables 33–40). These results accorded with the observation of domain*load interactions in the above mass-univariate analysis.

**Stage-general and stage-specific coding of WM domains**. We investigated the degree to which features of the domain-specific patterns of activation and FC were sustained across processing stages. We separated events by stage and estimated the domain classification performance distribution for a global model trained on a subset of data taken from all stages and tested against the replication set. The results showed that the regional activity and dFC patterns that encode WM domains classification generalised across stages, i.e. across stages a similar pattern differentiates domains (mean global accuracy = 76.34%, Fig. 4i. $p < 0.0001$, see Supplementary Tables 41, 42). We then examined whether domain states were 'stage specific', i.e., there are states that expressed predominantly during a specific processing stage and coded for the different domains. To test this, we trained domain classification models on a subset of data from each individual stage (i.e. Encode, Maintain and Probe) and tested these models on events from all stages from the held-out and replications sets (Fig. 4j).

Each stage-specific model has events that match the stage it was trained on as well as mismatch events. Therefore, we examined (for each set and metric) the averaged performance for match and mismatch events. In general match events

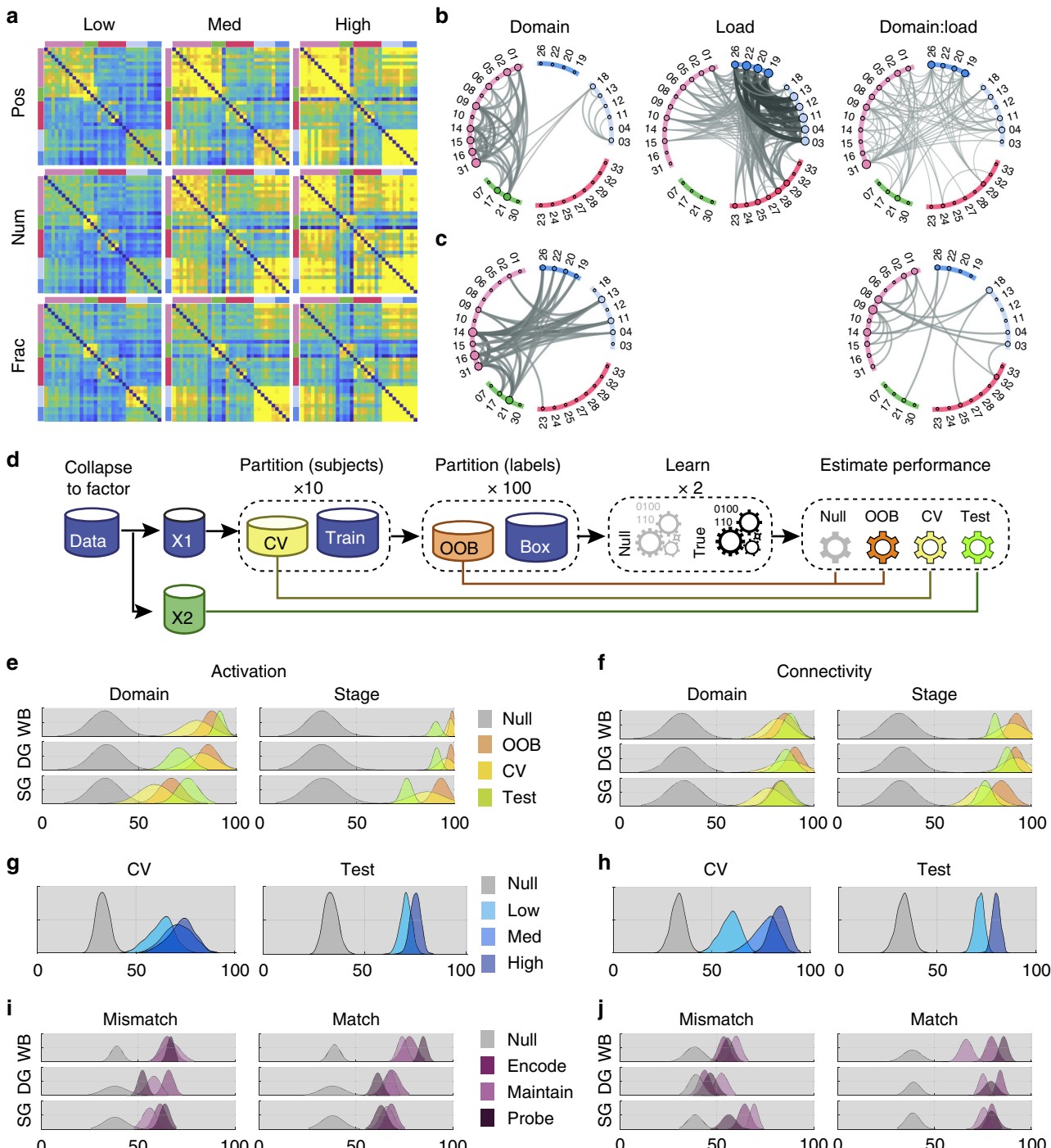

**Fig. 4** Decoding domains across experimental factors. **a** Mean functional connectivity matrices for the WM domains and loads collapsed across stages in Study 1. Visual inspection of these plots indicated upregulation of connectivity as a function of load. **b** Mass univariate analyses including data from both studies and collapsed across WM stage showed connections with significant main effects of load and domain, and significant load × domain interactions. Note, the significant effects of load centre on intra- and inter-connectivity of the visual ROIs. Repeating this analysis for the maintenance stage only (**c**) showed significant main effects of domain and domain × load interactions. Critically, when the effects of visual input were controlled in this manner, there were no main effects of load. Significant interactions centred on the intra and inter-connectivity of frontoparieal ROIs. (FDR corrected at $p < 0.01$). **d** Machine-learning pipeline. Data from Study 1 were partitioned into ten different cross-validation subsets. Training data were bootstrapped with replication 100 times to form training and validation sub-partitions. These data were used to form both true and null models across participants (i.e., scrambling the training labels). All data from Study 2 were assessed across studies. performance histograms for both activity (**e**) and connectivity (**f**) for decoding domains, and stages showed high accuracy for real (yellow, green and orange) vs null (grey) models. **g**, **h** Classification of domains based on DG ROI activation (**g**) and connectivity (**h**) was significantly better at high relative to low WM load. **i**, **j** Domain classification models trained using events from all stages generalised well for both activity and functional connectivity. Notably though, training models using a specific stage a significant reduction in classification accuracy when testing events from stages that mismatched the model for connectivity but not activity. This accorded with some stage-specific coding of domains in the dynamic connectivity state of the network

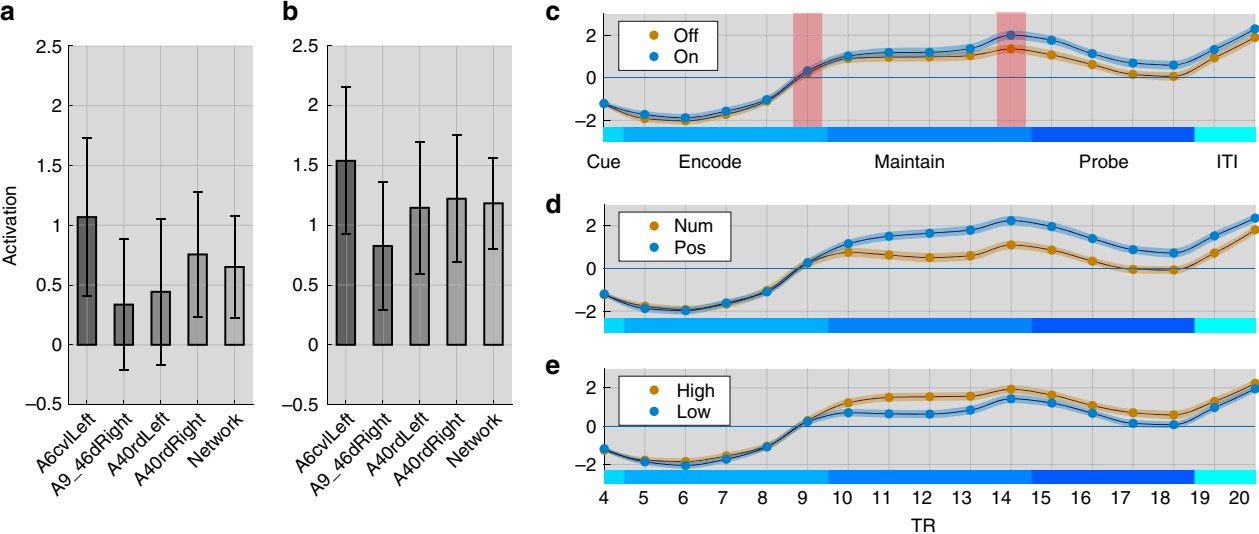

**Fig. 5** Time series analyses. **a** ROI activity at the end of the encoding phase (TR9) for the manipulation minus maintenance trials. There were no significant differences in activity for these conditions prior to the retro-cue. Error bars represent standard error of the mean. **b** ROI activity just prior to the probe (TR14). There were significant increases in activity for the manipulation vs. maintenance trials in all ROIs. **c–e** Plotting the entire BOLD time series averaged across all four ROIs showed that in addition to the effects of manipulation there were also effects of both domain and load. Therefore, the set of brain regions associated with manipulation demands is also sensitive to other aspects of WM

exhibited significantly (Supplementary Table 43) higher accuracy ($Acc_{match} = 74.16\%$) than the mismatch counterparts ($Acc_{mismatch} = 58.74\%$). Interestingly, activation-based models the differences between match and mismatch ($Acc_{match} = 70.1\%$, $Acc_{mismatch} = 61.76\%$) were less pronounced than for models trained using connectivity ($Acc_{match} = 78.2\%$, $Acc_{mismatch} = 58.75$). Importantly, in both the DG and SG sets, stage-specific accuracy significantly out-performed global accuracy ($t^{DG}_{(18)} = 6.14$, $t^{SG}_{(18)} = 7.02$, $p < 0.0001$). These results demonstrate that domain × stage interactions are coded in network connectivity as opposed to activity. It also indicates a particularly prominent role for frontoparietal-visual connections in the coding of domain × stage conjunctions.

**Contrasting manipulation vs. maintenance.** Several studies from the classic literature reported a dissociation between lateral frontal lobe areas involved in the maintenance and manipulation of WM items[2,14]. We tested the classifiability of maintenance and manipulation events from Study 2 using a liberal within-study cross-validation pipeline (i.e. leave three subjects out 10 times and estimate classification accuracy based on the mean distribution of this subset). Classification accuracy was not significantly higher than the permutation null distribution for models trained on activation patterns or FC patterns for the DG, SG or WB ROI sets (Supplementary Tables 44, 45). Notably, repeating the same within-study pipeline to WM stages and WM visual domains generated highly significant results. This null-finding was unexpected; therefore, we conducted a timecourse analyses focusing at finer temporal grain on the activation timecourses of four 5 mm radius spherical ROIs placed at the peak dorsolateral frontoparietal coordinates from a classic study of WM manipulation[43]. Analysing the difference in activity at the end of the manipulation vs. maintenance periods (where the haemodynamic response should be close to peak) showed the expected manipulation-related increase in activation ($t = 3.93$ $p = 0.001$) when averaged across all four ROIs (Fig. 5a, b). Repeating this analysis for individual ROIs showed the same effect in all cases (left parietal $t = 2.75$ $p = 0.015$; right parietal $t = 2.63$ $p = 0.019$; left frontal $t = 3.15$ $p = 0.007$; right frontal $t = 2.17$ $p = 0.047$). In contrast,

there were no significant effects at the end of the encoding period (all $p > 0.3$), i.e., just prior to the retro-cue. Plotting the full FIR timecourse for each condition showed that these effects of manipulation were overshadowed by larger effects of load and domain (Fig. 5c–e). This result accords with the view that the brain regions that are most closely associated with manipulation demands also have broader roles in WM[43].

**Characterising spatial topologies with sparse models.** We inspected the features that contributed most prominently to classification of WM domains to compare them with predictions from the classic localist literature. Regularised sparse models were trained[44] with whole-trial data from the DG and WB ROIs. This approach estimates the limited set of connections required for accurate classification of each domain along with a weighting matrix for those connections and a bias term (Fig. 6a, d). Notably, in both ROI sets the sparse models performed as well as the dense models. Back projecting the feature weightings for number WM rendered a left lateralised network, including connections spanning visual, parietal, temporal and posterior frontal ROIs. Spatial WM, included a dorsal network spanning inferior parietal cortex, motor cortex and posterior/dorsal frontal cortex ROIs. Fractal WM included right lateralised connections between the ventral visual association areas and projecting between visual, parietal and more ventral frontal cortex ROIs. The overall topographies of these projections were consistent across ROI sets albeit comprised of different connections and critically, they were comparable to those expected based on the classic literature. Nonetheless, further inspection (Fig. 6c, f) of the effect of WM maintenance relative to the ITI showed that none of the connections were exclusively active for just one domain. Instead, they were generally active for all domains relative to the resting ISI, but most strongly for different domains. Thus, WM domains mapped to densely overlapping patterns of connections as opposed to mutually exclusive networks.

## Discussion
Our results provide converging evidence to support the network-coding perspective on human WM function (Fig. 7).

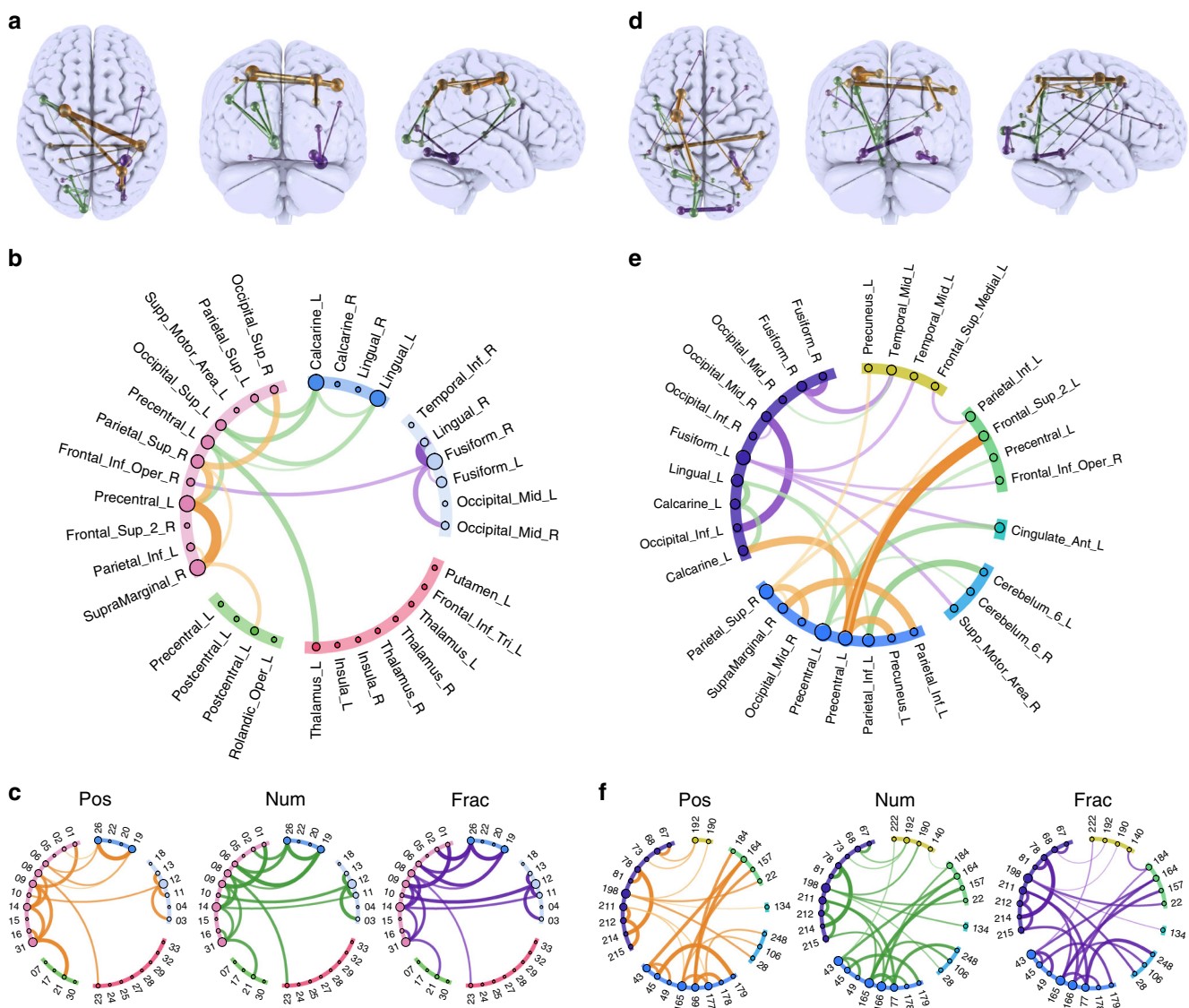

**Fig. 6** Sparse model's anatomical projection. Projections (**a**) and schemaballs (**b**) of weightings for the sparsified classification model of stimulus domain, generated from whole-trial data of the DG ROI set. GLMnet identified the minimal subset of connections required for accurate classification of each stimulus domain along with weightings and bias terms. Lines represent the weightings of connections for each WM domains as follows: orange = position, green = number and purple = fractal. **c** Schemaball of *t*-values for maintenance of each domain relative to the ISI. Thresholded at $p < 0.05$ with FDR correction for multiple comparisons. Note, none of these connections, which contributed the most information for classification of the stimulus domains, was active for a single stimulus domain exclusively. **d** Repeating the sparsification analysis with the whole-brain ROI set generated comparable results in terms of the gross topography of position, number and fractal connectivity patterns (**e**). **f** As per the DG ROI set, the connections within the whole-brain that contributed the most information for classification of domains were generally active for multiple domains during maintenance

Behaviourally distinct aspects of WM including stimulus domains and processing stages evoked strongly dissociable multivariate patterns of activity and connectivity across heavily overlapping brain networks. These aspects of WM were classifiable with remarkably high accuracy even when analysing the SG ROIs, which had the broadest roles as they were active across all conditions. Furthermore, the finer grained domain by stage conjunctions were coded by connectivity but not activity patterns. Moreover, the effects of maintenance load on network connectivity were characterised by heightened classifiability of stimulus domains, and at the mass-univariate level load by domain interactions as opposed to load main effects. This accords with a fine-tuning of the network towards a domain-optimised state when the difficulty of the WM task increased. Critically, these task-evoked brain states were consistent across participants and

independent studies. Therefore, they reflect on a fundamental level how human brain networks are organised to flexibly support diverse WM demands.

On the surface, the notion of a dynamic network coding mechanism can seem incompatible with early localist models of the brain-imaging and neuropsychological fields. This is problematic because much of the neuroimaging field is dominated by localist mappings; indeed, in some cases they have been replicated across multiple studies and groups[16]. The central aim of our study was to determine whether the localist and network frameworks could be reconciled. Based on our results, we argue that this is clearly achievable. More specifically, different conclusions have been drawn from our and previous studies because of two main factors. First, previous studies have typically used WM tasks that are designed to contrast limited numbers of WM conditions.

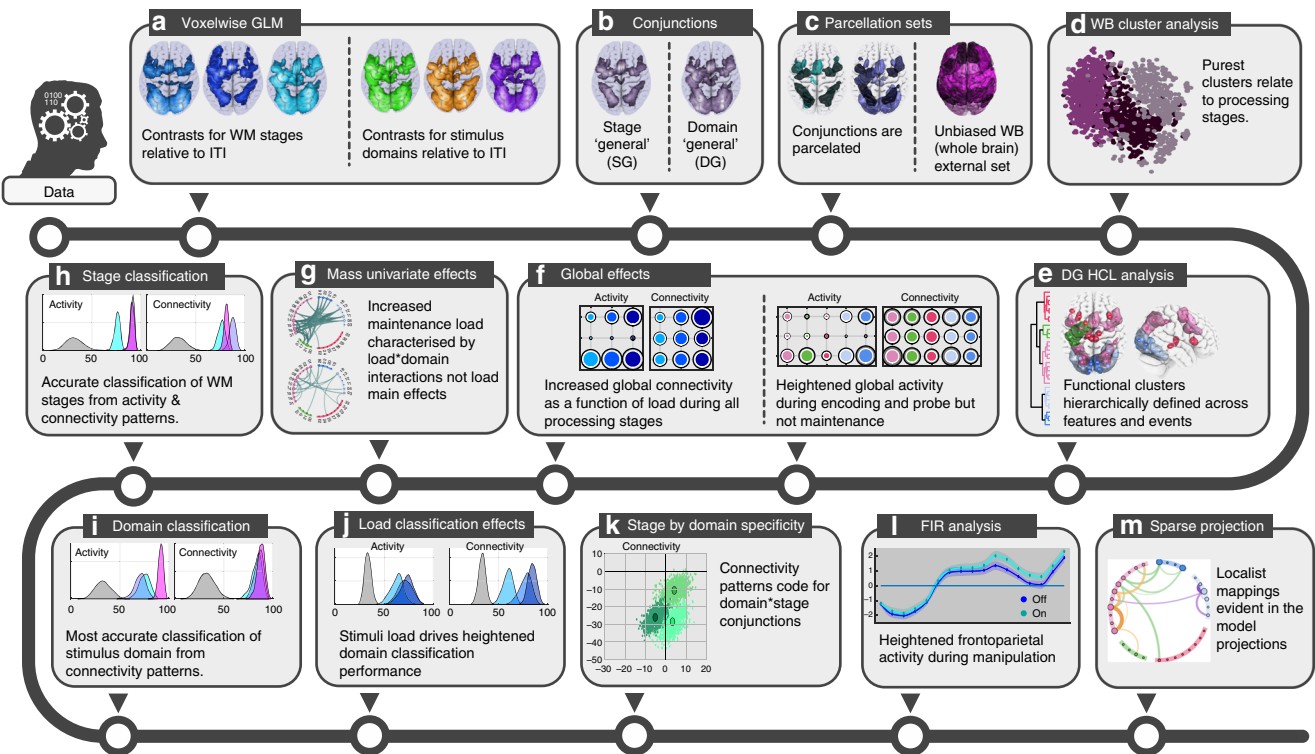

**Fig. 7** Effects overview. **a** Voxelwise statistical parametric maps were calculated with general linear models for processing stages (in blue) and stimulus domains (number, spatial and fractal pattern). **b** Conjunctions were calculated between the statistical maps to generate DG and SG maps. These accorded with previously reported Multiple Demand Cortex. **c** Conjunction maps were sub-divided into discrete ROIs using a 3D watershed algorithm. **d** Hierarchical clustering showed that the purest stratification of ROI activation patterns related to WM processing stages as opposed to stimulus domains. **e** Hierarchical clustering also generated interpretable functional clusters of DG ROIs. **f** Effects of task conditions on ROI clusters showed heightened global activity and connectivity during encoding, and heightened connectivity only during maintenance. **g** Mass univariate analyses showed increased maintenance load was characterised by domain-load interactions as opposed to load main effects, with these being focused on frontoparietal connectivity. **h** WM stages were classified with high accuracy from ROI activity and connectivity patterns. **i** Stimulus domain was classified but with superior accuracy for connectivity patterns. **j** Stimulus domain classification was superior at heightened load. **k** Classification of domains generalised across WM stages but was superior within stage when based on connectivity patterns, indicating coding of domain × stage conjunctions. **l** Heightened activation for WM manipulation was evident within the expected ROI set; however, the same ROIs showed greater sensitivity to WM domains and stages. **m** Anatomical projections of sparse classification models corresponded well with the mappings expected based on the classic localist literature

For example, contrasting WM for object vs. spatial stimuli[45], encoding vs. maintenance[46], maintenance vs. manipulation[43], or low vs. high load[47]. Such designs when applied in unconstrained whole-brain analyses give the illusion of one-to-many functional-anatomical mappings, i.e., where discrete circuits or processing streams are dedicated to the WM function that is being studied. When coupled with hypothesis driven analyses that focus on limited volumes of the brain, for example, the lateral frontal cortices, this problem is exacerbated because it leads to the illusion of one-to-one mappings, where a single functional anatomical module is exclusively ascribed the function of interest (see refs. [20,25,26] for reviews).

Here, we applied a more holistic analysis approach, designing multifactorial tasks that enabled a broad range of behavioural conditions to be contrasted[25,48], and using data-driven analyses that were not biased by assumptions regarding how those conditions map onto the brain. Notably, in our dataset, conducting simple subtractive contrasts between any given pair of conditions in isolation would have produced the illusion of mutually exclusive or modular brain systems; conversely, in the context of the broader findings these mappings were shown to be many-to-many as opposed to mutually exclusive. Given the broad cognitive scope and lack of spatial constraints in our study, we find it reassuring that the findings reported in the localist literature were not exposed as incorrect *per se*; instead, in all cases they were

recast as salient features of the densely distributed heavily overlapping multivariate activation and connectivity patterns that characterise different aspects of WM.

Prominent incongruences between localist models were also reconciled when recast within the network framework. For example, a debate of the early literature regarded whether dorsal and ventral modules within the lateral frontal cortices had specialised roles in visual and spatial WM[6], an extension of the putative 'what vs. where' visual processing streams[49], or instead were responsible for applying-specific processes to information of any type, e.g., maintaining/monitoring vs. manipulating information in WM[2,3,14,43]. When taking a multivariate view, our analyses show evidence for both mappings during the performance of the same task. Trials where spatial locations were encoded and maintained could be classified based on increased connectivity through a dorsal network dominated by visual, parietal and posterior frontal areas. Fractal trials were classified based on increased connectivity throughout a ventral network spanning visual processing streams and projecting to more inferior frontal areas. Critically though, these connections were not exclusively sensitive to either stimulus domain. Furthermore, the set of frontal and parietal brain regions associated with manipulation demands was confirmed to be more active during manipulation vs. maintenance trials, yet, all four of those ROIs were also sensitive to stimulus domain and the level of

maintenance WM load. Therefore, as opposed to there being discrete systems for these WM demands, the same network of brain regions dynamically reconfigures its detailed pattern of activity and connectivity dependent on them.

One might argue that this general involvement of brain regions in diverse WM demands favours globalist perspectives. which based on the observation of common activation patterns across diverse cognitive tasks proposed the existence of a central and highly flexible cognitive resource in the brain, i.e., Multiple Demand Cortex[32,33]. Again, the modular and globalist perspectives can appear incompatible; however, they were reconciled within the network-coding framework; the domain general perspective was correct insofar as brain regions corresponding to Multiple Demand Cortex were commonly recruited across all WM domains and stages. However, this volume was not comprised of functionally homogeneous brain regions or even a simple binary sub-division into sub-networks[48]. Instead, the ROIs had diverse profiles in terms of the combination of WM conditions under which they were most active; this conforms to a many-to-many mapping between function and anatomy. Indeed, all the above aspects of WM were classifiable from the multivariate pattern of activation and connectivity even within the most broadly active subset of brain region (SG). From this, we can infer that Multiple Demand Cortex dynamically codes for different task demands in a multivariate manner not only at the neuronal[50] and voxelwise level as previously reported[51,52], but also at the macroscopic level through transient changes in its internal network configuration[53].

In summary, the network coding perspective provides an overarching framework that can reconcile insights from diverse studies of WM. We believe that the distributed coding of different aspects of WM that is evident across macroscopic brain networks is analogous to that which is observed in the coding mechanisms of local populations of neurons[54]. Such coding likely is a scale-free property of how networks efficiently and robustly support diverse information types and processes[50]. Here, using multivariate analysis methods that align with these distributed coding mechanisms provided remarkably high classification accuracies with models that generalised across individuals and independent studies. Tentatively, we posit that the capacity to transiently express brain states that are optimal for performing-specific tasks may underlie population differences in cognitive ability. Given the reliability of our classification approach, we believe that univariate measures summarising the strengths of expression of the connectivity patterns for stimulus domains and processes should be explored in studies of group and individual differences in WM and as outcome measures in trials that seek to enhance WM functions. The latter application is particularly promising given recent studies showing that the functional connectivity of the WM network is amenable to modulation by non-invasive brain stimulation[55] and augmentation by focused cognitive training[56,57]. Future work should also focus on analysing directed information flow using effective connectivity methods to determine how transitions between these task-evoked processing states are orchestrated within the human brain[29,58,59].

## Methods

**Experimental design and data collection**. Participants— 19 right-handed healthy adults (6 female, mean age 22.368, range 18–28 years of age) participated in Study 1 and 17 right-handed healthy adults (11 female, mean age 23.938, range 19–41 years of age) participated in Study 2. All participants had normal hearing and corrected to normal vision. Before commencing the study they read instructions regarding the task and protocol, agreed to experimental procedures and underwent a short training session to ensure that they could perform the task. The training session consisted of approximately 15–20 min practicing the task on a laptop. They then entered the MRI scanner and undertook the task. One participants in Study 2, was excluded due to a technical error that resulted in corrupted behavioural recordings.

Sample sizes were selected to align with studies of WM in prior literature. The studies were approved by the University of Western Ontario ethics committee.

Procedure— In both studies participants performed three runs of similar cognitive experiments designed to manipulate WM domains and processing stages. Each run was composed of pseudo-randomly allocated blocks, where each block included a single trial with distinct encoding, maintenance and probe stages (Fig. 1).

Study 1— Participants were required to encode and maintain a set of features from an array of compound stimuli, composed of a pseudo-randomly selected numbers and fractals placed at random spatial positions within a 4 × 4 grid. Each trial began with a pre-encoding cue directing participants to focus on features from one of these three stimulus domains (number, fractal or spatial). Then, three, five or seven compound stimuli were presented within the 4 × 4 grid. After 10 s of encoding, stimuli were removed, and participants were required to maintain the features from the cued domain for 10 s. Subsequently, participants were presented with a probe array where one each of the numbers, fractals and locations had been shuffled. They were required to indicate within a 10 s timeframe the location of the shuffled item that was within the currently maintained domain. The trial terminated at the point in time when the participant responded and the next trial began after a 10-s inter-trial-interval (ITI), during which a fixation cross was displayed and they were instructed to rest. In the imaging analysis, this (ITI) provided the baseline for comparing all other events, i.e., where no overt processing of the WM task was required.

Study 2— Participants were again required to encode and maintain a set of features from an array of compound stimuli in a similar to the first in design; except for three major differences. (1) A second retro-cue was added to cue the participants to either manipulate (add 3 or mentally rotate 90 clockwise the spatial pattern) or maintain the encoded information. (2) Stimulus domains included numbers and spatial locations only. This reduced complexity in the factorial design to compensate for the addition of the manipulation condition. (3) Similarly, the load condition were limited to two levels, these being three or six items per stimulus domain.

Behavioural data collection— In both experiments, data were collected in three runs of scanning acquisition. In the first experiment, each run contained 18 trials, two each from nine possible combinations of stimulus domains (number, fractal and spatial) and WM load (3, 5 and 7). In the second experiment, each run contained 16 trials, two each from eight possible combinations of cognitive processes (Maintenance and Manipulation), stimulus domains (number and spatial) and WM load (3 and 6). Stimuli were presented on a back-projection screen visible from the bore of the MRI scanner via a mirror mounted to the head-coil. Responses were taken with a custom MRI-compatible trackball mouse. Both WM paradigms were programmed using Adobe Flash Builder 4.5 and embedded in a scanner interface programmed in Visual Basic.

MRI data collection— Brain images were collected using a 3 Tesla Trio TIM SIEMENS Scanner. A T2 weighted echo planar image depicting blood oxygenation level dependent (BOLD) contrast was acquired every 2000 ms. The first ten images were discarded to account for equilibrium effects. Images consisted of 36 3 mm slices, with an 80 × 80 matrix, 240 × 240 mm field of view, 30 ms TE, 2 s TR, 45° flip angle, 2.65 ms echo spacing. A 1 mm resolution MPRAGE structural scan was also collected for each with a 256 × 240 × 384 matrix, 900 ms TI, 2.98 ms TE and 9° flip angle. Acquisition in Study 2 was almost identical except that the flip angle was 60°.

Imaging Quality control— Raw bold signal to noise ratio (SNR) metrics were extracted across studies using an in-house implementation of the metrics proposed by Friedman[60]. A multivariate outlier detection analysis was performed to identify any low SNR values. No scans were omitted at this stage.

**Exploratory analysis**. Pre-processing— Before analysis, data were preprocessed using a custom pipeline that included functions from SPM12 (Statistical Parametric Mapping Welcome Department of Imaging Neuroscience), FSL (FMRIB Software Library v5.0) and MATLAB 2017b. Specifically, data were slice-timing and motion corrected, spatially warped onto the standard Montreal Neurological Institute template using a custom DARTEL template generated from all the structural scans (for each study independently), and spatially smoothed with an 8 mm³ full width at half maximum Gaussian kernel. The data were high-pass filtered (cutoff period of half of the scan length—approx. 300–400 s) to remove low-frequency drifts in the MRI signal.

Whole-brain BOLD activity estimate— Nuisance experimental matrices were formed using motion estimates (in an extended 24-parameter model) and motion outlier (spikes) events based on Yan et al. recommendations[61]. Then a single subject design matrix was constructed using the HRF convolved experimental onsets and nuisance matrix. This was applied to the fMRI stack to estimate beta coefficient fits using the classic mass-univariate GLM in SPM12. Finally, whole-brain maps depicting statistical parametric estimates were generated using predefined contrasts of interest. In the voxelwise analysis, this was followed by 2nd level group contrasts against the null distribution, focusing on the following experimental dimensions across conditions: (1) visual domains (numbers, fractal and spatial), (2) WM stages (encode, maintain, probe), (3) load (high, medium, low) & (4) and manipulation (on, off). In Study 1 there was no 'manipulation' condition. In Study 2, the visual domains comprised only numbers and spatial stimuli and there was no medium load.

Data structures— Single subject general linear models capturing regional changes in activity were constructed using a mini-block design, in which each stage (e.g., cue, encode, maintain, probe) of each individual trial was captured by a separate predictor, alongside nuisance covariates. The resultant whole-brain maps depicting parameter estimates were taken to the group level, creating stacks composed of $19 \times 3 \times 3 \times 3 \times 4 \times 2 = 4104$ events (i.e. subjects × runs × domains × loads × stages × repeats) for Study 1 and $16 \times 3 \times 2 \times 2 \times 2 \times 2 = 3072$ events (i.e. subjects × runs × manipulation × domains × loads × stages × repeats) for Study 2.

Conjunction analyses— We used minimum statistic conjunction-null tests[36,62] to examine intersections across experimental domains as per SPM12's suggested flow. Specifically, we used the minimum T-statistic over n orthogonal contrasts. Inference was based on individual cluster corrected statistical volumes per contrasts, which were later combined using a logical AND. We derived minimal cluster size by performing uncorrected analysis with a relaxed threshold ($p < 0.01$), then applying the minimal false-discovery-rate (FDR) cluster ($P_{FDR} < 0.05$) to generate a cluster corrected map for each of the contrasts in the specific conjunction. This enabled us to infer a conjunction effect at significant voxels. The above procedure was performed in an identical manner across both studies and the following comparisons were performed: (1) Domain General (DG) a 3-way conjunction over the three stimulus domains in Study 1 and independently on the two visual domains in Study 2; and (2) Stage General (SG) intersecting the three processing stages in both studies (i.e. encode, maintain and probe).

Similarity calculation— Minimum statistic conjunction-null tests were performed separately for each study[36] in SPM12 with voxelwise thresholding at $p < 0.01$ followed by false-discovery-rate (FDR) cluster correction across the whole-brain mass at $p < 0.05$. The quotient of similarity (QS) summarized by dice coefficients was calculated across the two studies to identify clusters that were reproducibly identified for the DG and SG conjunctions across both studies. Specifically, binary masks were created from the cluster thresholded maps and the similarities calculated using the following formula:

$$QS(A, B) = \frac{2(A \cap B)}{(A + B)}. \qquad (1)$$

Data driven parcellation— Our analyses focused on ROI's produced from the above conjunction maps and a whole-brain parcellation made available by another group[63]. These ROI's enabled both dimension reduction of the voxelwise space and calculation of task-evoked connectivity matrices. Thus, they form a basis for connectomics[54].

Data driven parcellation— We used a "watershed transform"[64], to segment the inverse of the minimal conjunction maps into ROIs. This common segmentation procedure treats the three-dimensional statistical volume as a multi-dimensional surface where high and low intensities represent elevations. The algorithm iteratively 'filled' independent catchment basins (CB) with unique labels by flooding the various-independent local minima in the statistical volume and their surroundings. At the end of the segmentation, we filtered out small ROIs, by applying a >50-voxel exclusion criteria. The conjunction maps from Study 1 were parcellated into ROIs using our in-house 3D implementation of the watershed algorithm. Using the ROI set a database was formed containing the mean parameter estimates for each ROI and each WM event within the stack (as defined above).

Hierarchical clustering— Focusing on the (more inclusive) DG-ROI set we collapsed events to compare visual domains and processing stages (i.e. averaged across runs and loads). This formed an $m \times n$ matrix (one for each experiment), where $m$ are events averaged for each individual (subjects × visual domains × processing stages) and $n$ are the ROI's, e.g., from DG-conjunction. Using Euclidean distance and the Ward method[65] (which minimizes the total within-cluster variance) we clustered the ROIs and WM conditions along both axes. We used agglomerative hierarchical clustering[40] (HCL) to uncover structure in the hierarchies of the activation data stack defined above. Dendrogram Inconsistency was used to define the natural data-driven cutoff across the HCL hierarchy. This threshold was measured as the difference between height of the current link and the mean height of its sub-graph (depth = 5) divided by the standard deviation of the sub-graph.

Cluster purity measure— Purity is a common external evaluation criterion of cluster quality. As the clustering is performed in an unsupervised manner, and we should not expect correspondence between number of classes nor class assignments due to chance alone, we matched each cluster to the class that was most frequent and measured the intersection between them. Accuracy was measured as the percent of total events that matched this initial pairing. Where $N$ is the number of total events, $k$ the number of clusters, $\Omega = (\omega_1, \omega_2, ..., \omega_k)$ the set of clusters and $\mathbb{C} = (c_1, c_2, ..., c_k)$ the set of classes.

$$\text{Purity}(\Omega, \mathbb{C}) = \frac{1}{N} \sum_{i=1}^{k} \max_j |\omega_i \cap c_j|. \qquad (2)$$
$$(\Omega \in \mathbb{N}, \mathbb{C} \in \mathbb{N})$$

Transient functional connectivity estimate— While there exist various methods to estimate connectivity between two regions[66] dynamic or model-based connectivity measures pose a challenge[67]. Here we used psycho-physiological interaction models[28,30] (PPI), which estimate the task-evoked functional connectivity between each pair of brain regions. We used this approach primarily for its simplicity and computational efficacy. Notably, the classic method is limited to a single-PPI contrast. Recently a generalised form of PPI was suggested to assess simultaneously multiple dimensions of the experimental space. We used a custom MATLAB implementation of the following model:

$$Y^T = \beta_0 + [Y^S, H(X), E]\beta_G + [Y^S \times H(X)]\beta_j + e, \qquad (3)$$

where $X$ was, the matrix containing psychological timecourses (i.e. timecourses for encode, maintain and probe events) and $H(X)$ was the HRF convolution of that matrix. $Y^T$ was the target time series and $Y^S$ the source time series. $E$ was the nuisance regressor matrix defined previously in the pre-processing stage. $\beta_G$ included weights of no-interest and $\beta_j$ the weights for the PPI predictors, which were the target of further analysis. $\beta_0$ was the intercept and $e$ the residual error. This model was estimated for all pairs of connections to form a connectivity matrix and upper lower triangles were averaged to form an undirected weighted connectivity matrix for each condition in the design matrix.

**Machine learning**. Multi-class classification— Multivariate classification based on fMRI measures has been successfully applied extensively in the past, e.g., to explore multivariate discriminant neuronal coding relating to stimuli, task rules, emotions and more[52,68,69]. Although it has mostly been used in the context of local voxel patterns, combining this approach with global patterns extracted using some high level parcellation scheme allows in-depth inspection of patterns from both activity and connectivity across the brain. In this paper, we used dense and sparse multi-class classification models in a 'one-vs-all' scheme which means that a model was trained to differentiate between a class and all events that are not of that class. The classification accuracy was an estimation of the information content in a dataset[70]. To achieve this, a stack of models per class was constructed. For the dense models we used an 'Error Correcting Output Codes'[71] (ECOC) ensemble approach to solve the multi-class problem. We used the MATLAB built in function with linear support vector machine binary classification as the actual learning algorithm to discriminate between groups. For the sparse models we use multinomial logistic regression by Friedman et al.[44]. Using the GLMnet MATLAB toolbox https://web. stanford.edu/~hastie/glmnet/glmnet_alpha.html. This identifies the minimal set of mutually exclusive features for each class along with weighting matrices and class biases.

Null-model generation— To estimate significance relative to the null distribution, we randomly divided events from Study 1 into two unequal parts (75/25%) based on the participant's id. The smaller sample was our within-study validation set, and the bigger was the training set. We also included all the data from Study 2 as an independent test set. We trained two classification models one using the actual labels from the training set and another model (the null model) using shuffled labels (i.e. destroying any possible connection between dependent and independent measures). We then estimated the model's performance within study (using the validation set), and the models robustness across studies (using the test set). This step was repeated a 1000 times to form two distributions, our actual performance that quantifies the relevance of the neuronal information to the classification task and the null distribution that quantifies the classification of events by chance.

Feature selection flow— Neuroimaging data can range from highly selective hypothesis driven features (i.e., specific regions of interest) to whole-brain voxelwise inspection (several to tens of thousands of features). Connectivity data provides an almost squared increase in dimensionality complexity relative to regional activity (from tens to billions of features). This raises two complementary problems, multicollinearity, where multiple features share similar relevant information for classification and the "curse of dimensionality", where the number of features far surpasses the number of observations. Dealing with the latter is possible by using sparse logistic regression instead of SVM.

Lambda selection— We trained our sparse models with a range of regularization values using an exponential decay function. Optimal lambda was selected from this range by finding the maximum absolute second derivative over the sum minimal cross entropy values derived from the training set (also known as finding the knee in scree plots).

Performance measures— In the context of this work, accuracy was defined as the balance between the precision and recall (see detailed definitions Sokolova et al.[72]), commonly known as F1-score, which is considered a more appropriate measure for multi-class classification problems.

**Statistical significance testing**. To determine statistical significance, we used the methods detailed below. All testing was performed using MATLAB 2017b statistical toolbox and costume MATLAB functions (see code availability).

Behavioural significance testing— We used repeated-measures ANOVA to examine behavioural effects for both accuracy and response time. In the latter case we used log transformed response times to better comply with the normality assumption of the test. When applicable, this was followed by a post hoc multi-comparison pairwise tests to identify the basis and direction of the significant effect.

Cluster averaged significance testing— We also used repeated-measures ANOVA to examine global effects in the domain general ROI set for clusters

(defined in the clustering section) averaged activation and connectivity. In the later case, we distinguished between inter and intra connectivity. Mauchly's test for sphericity was performed to assess whether ANOVA assumptions were violated and if so Greenhouse Geisser corrected $p$-values were used. When applicable, this was followed by a post hoc multi-comparison pairwise test to identify the basis and direction of significant effects. All post hoc tests used Bonferroni correction.

Classification accuracy significance testing— Following[40,73–75] to estimate classification accuracy significance we used a conservative repeated cross-validation approach to form paired performance estimations as defined in performance estimation section. We then estimated the empirical probability as $p = \frac{b+1}{m+1}$, where $b$ was the number of events where $Acc_{perm} > Acc_{obs}$, i.e. the number of events where the permuted null model out-performed the model trained by real data, and $m$ was the number of random sampling pairs (10 partitioning × 100 bootstraps). Effect size was then calculated using the Mann–Whitney U-test to compare between the permuted and observed performance (i.e., taking the conservative-independent replication set values).

Measure comparison significance testing— To compare between activation and connectivity accuracies we averaged performance within each of the ten-stratified partitioning (selecting ten different sets of training and testing based on unique subject's id's see above) and used $t$-tests to compare distributions above and beyond the different sets.

Match/mismatch significance testing— To address the extent to which features in the models for classification of domains generalised vs. were specific to WM stages, we trained stage specific domain models using events from each processing stage (encode, maintain & probe) independently. These were then applied to data to which they were naive from all three domains and all three stages.

Domain by load thresholded network analysis— To uncover-specific connections that were upregulated as a function of both load and domain (i.e. interaction effects) we applied linear mixed effects models (using MATLAB *fitglme* function) independently for each connection. Specifically, the upper triangle of the connectivity matrix was extracted for all trial events, and for each connection the following model was fitted '$Y = 1 + study + domain \times load + (1|subject)$'. ANOVA was then used on the model to assess main and interaction effects. $P$-values were FDR corrected for multiple comparisons. This procedure was performed first for whole trials (i.e. averaging across processing stages) and then for maintenance only.

**Code availability**. All custom MATLAB routines and data used to generate the analysis and figures of this paper will be committed upon acceptance for review to the following GitHub: https://github.com/esoreq/WM.

## Data availability

Neural and behavioural datasets for both studies have been made available online at OpenNeuro Study one; https://doi.org/10.18112/openneuro.ds001634.v1.0.1. Study two; https://doi.org/10.18112/openneuro.ds001635.v1.0.1.

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

## Acknowledgements

E.S. undertook this work as part of his Ph.D., funded by Imperial College London and completed it while employed on MRC project grant MR/R005370/1 awarded to R.L. and A.H.

## Author contributions

All three authors conceived the study. A.H. designed and programmed both tasks, contributed to the design of the analysis pipeline and helped write the article. R.L. contributed to the design of the analysis pipeline and helped write the article. E.S. designed and developed all components of the analysis pipeline, performed all analyses, created the figures and authored the article.

## Additional information

**Competing interests:** The authors declare no competing interests.

