## [Peer Review File · Nature Communications]

Reviewers' comments:

Reviewer #1 (Remarks to the Author):

"Dynamic network coding of working-memory domains and working-memory processes," Soreq, Leech, and Hampshire. This manuscript describes the results from several multivariate and connectivity-related analyses of fMRI data from two sets of working memory tasks, different subjects performing each, with results from one experiment validated against the results from the other. The idea is to identify and characterize high-dimensional brain states that underlie different WM processes (encoding, maintenance/manipulation, retrieval) crossed with different stimulus domains (spatial, object, digit). This is an important undertaking, and one that, a priori, should yield valuable information about large-scale dynamics of the neural bases of WM function. I had difficulty, however, with what reads like a recitation of ad hoc interpretations of results that, albeit generated by sophisticated methods, don't leave me with a better understanding of WM than what I brought to the manuscript. Additionally, the authors report several puzzling failures to replicate well-established findings in the field. It should be the case that, unless what the authors characterize as "early neuropsychological perspectives" are outright wrong (and this would mean that, in some cases, results from tens or more different groups around the world have been wrong), the dynamical network-level methods brought to bear on this dataset could show that the "early", often univariate, results hold, but perhaps their interpretation changes, or they can be shown to be superficial byproducts of more fundamental processes that are captured by these more sophisticated analyses. Absent this, when an approach like the one here is carried out in the context of a failure to replicate classical results, the reader is left uncertain about the validity of the new results. (An analogy might be to, say, predictive coding analyses of visual perception. Although activity in V1 may be reinterpreted in terms of the feedforward propagation of error signals, and the feedback of information about updated priors, if these claims were made in the context of a study that failed to find classical orientation tuning in V1 neurons, the predictive coding claims wouldn't be taken seriously. What IS persuasive, in contrast, is if one can show the classical pattern, but then apply the fancy new ideas and analyses to argue that this activity is better understood in terms of a predictive coding framework, rather than the classical feature detection hierarchical model.)

Both of these concerns are seen, for example, in the section on "Load related changes in activity and connectivity as a function of stage." It is stated that "In accordance with our prediction, within the DG ROIs, there was no significant increase in activation during maintenance at higher load." In fact, however, there is no place in the Introduction, or elsewhere, in which this prediction had been articulated. This and comparable subsections of the Results make it read like an unprincipled, ad hoc recitation of findings that later get interpreted in what feels like a post hoc interpretation that's little more than a series of generalities that few would disagree with, independent of this paper. (E.g., "a highly adaptable network of brain regions supports diverse WM demands through a repertoire of connectivity states.")

The second concern is the failure to see a load effect. Indeed, on the day that I'm writing this review the April issue of Cerebral Cortex appeared in my inbox, and it contains a paper by Paul Bays in which he reanalyzes load effects reported in two already-classic papers from 2004: the Todd and Marois finding from IPS, with fMRI; and the Vogel and Machizawa introduction of the CDA (with EEG). The Bays paper cites at least ten replications (pre- and post-2004) of the basic fMRI finding, and I'd wager that there have been more than 100 replications of the CDA. Furthermore, some more recent studies that have applied multivariate analysis methods to data with load manipulations have reported that, as is the case with behavior, multivariate indices of the neural precision of stimulus information in WM declines monotonically with load (studies from Awh, Postle, Serences spring to mind). This is the opposite of what is reported here.) The same problem is true for maintenance vs. manipulation finding. Without an explanation for why the "classical" "first-order" results aren't duplicated here, the interpretability of the results of the more sophisticated analyses will be equivocal. In the case of "maintenance vs. manipulation," the

manipulation rule used here (rotate 90deg or add 3) are very easy, and so the absence of such an effect in these data may be indicative of the fact that this construct was poorly operationalized by this task, which would mean that the results can't be interpreted with reference to this construct.

Some of the broad generalizations don't accurately capture the current state of the field. For example, the very first sentence of the introduction is a statement that would have worked in the 1990s, but that an increasingly smaller number of scientists have subscribed to since that time. (Indeed, one of the two reviews cited argues for an explicitly contradicting position.) Another example is that the multiple demand (MD) system "would be predicted to show the least difference between [WM domain and WM process]. This is a mischaracterization, in that the point of the MD system is its ability to flexibly configure itself to accommodate whatever challenges an individual encounters, not unlike the "flexible coding" properties of pyramidal cells of the PFC.

The insula is not a subcortical structure!

Throughout the manuscript there is an overreliance on abbreviations that are nonstandard, idiosyncratic to this study, making it unnecessarily difficult to read (e.g., "WB" "SG" "DG" "FTP" "IPT"...)

Typos:

The text refers to "Extending a recent exploratory analysis of many functional anatomical mappings[24] in this journal;" but ref 24 is, in fact, a bioRxiv preprint.

"implantation"

Refs 3 and 5 have errors.

Reviewer #2 (Remarks to the Author):

Manuscript in-review: "Dynamic network coding of working-memory domains and working-memory processes", Soreq et al., 2018. Nature Communications.

Summary & Merits

The primary claim in this study is that localist frameworks for describing working memory (WM) (Baddeley, 2000; Fletcher & Henson, 2001; Owen et al., 1996; Roy, 2017) are insufficient, and in turn, a multivariate 'network coding mechanism' is more apt. Using data-driven dimensionality reduction methods to identify and assess WM networks by domain of processing, stage of processing, and demand on processing, followed by context-dependent psychophysiological interaction (gPPI), and multi-class pattern analysis, the results provide convergent support in this direction. It is noted that localist findings can be reconciled as an overly-sparse rendition of the network-space instantiated in WM. This allows for unification between studies (and generations of research with increasingly advanced acquisition and analytic techniques), as opposed to invalidating previous findings. This study highlights the importance of network science and machine learning in building theory about complex systems.

Major Considerations

1. The introduction requires further development. Specifically, the localist framework that this study is refining via network solutions should be delineated more thoroughly to give proper

context to the findings. Moreover, there is mention of a 'unifying framework' (page 2) that is not explicated or synthesized in the text. While one may discern that this framework is the 'network coding mechanism' mentioned throughout, it should be explicitly labeled as such. Indeed, a visualization or schematic that synthesizes the findings would be beneficial for a multi-dimensional study such as this (e.g., a visualization of what is touched upon in the discussion section).

2. There are various points throughout the analytical pipeline where alternate methods could be used. While the methods section and supplement do disclose the rationale behind many of these choices, such justifications should be recapitulated in the main text as well (salient examples: convolution method, functional connectivity estimation, cross-study dice coefficients, match/mismatch accuracies, and null-model generation). Moreover, there were instances when a methodological choice was either not adequately justified, or was novel and so required more interpretational detail.

3. Firstly, the use of data-driven conjunctions and parcellation schemes to derive the WM clusters that are either 'domain-general' (DG) or 'state-general' (SG) is not well justified against the alternative of a whole-brain, multi-modal approach (such as Glasser et al., 2016). Additionally, the use of a pre-defined ROI set can help avoid circularity in analyses (see [Kriegeskorte N, Simmons WK, Bellgowan PSF, Baker CI (2009) "Circular analysis in systems neuroscience: the dangers of double dipping". *Nat Neurosci.* 12:535-540. PMID: PMC2841687

<http://doi.org/10.1038/nn.2303>]). A predefined whole-brain ROI-set is used for a portion of the analyses, but it is not analyzed with the same level of detail as the DG or SG set. Additionally, the DG and SG sets suggest an intersection or collapse amongst domains and stages, respectively, however, they are later used in supervised learning algorithms that classify upon domain and stage (varied depending on the specific analysis). This is difficult to interpret, and requires more thorough consideration for the network coding mechanism. Alternatively, a whole-brain parcellation could be applied to the GLM data stacks (e.g., what is described in the "Exploratory analyses" portion of the methods section), and community structure (via HCL or maximized modularity) could be derived, naive of task condition (e.g., what is presumably utilized as the WB ROI set).

4. Secondly, the term 'rest' is used in interpreting the results of figure 3, but it is not clear how this is defined. The text suggests that the rest period is when the mean activity in visual clusters has a beta-weight of zero, which occurs during maintenance (also called 'delay' in some figures). This is then used to compare against frontoparietal and visual regions, for both activation and connectivity during encoding and delay. In the case of contrasting visual-to-visual, it is thus implied that comparing delay to rest is a self-comparison, although this is not clear. In the case of contrasting frontoparietal-to-visual, this raises the question of why a visual-system-determined baseline would be used (as opposed to a FPT baseline)? Regarding the same figure (3c), there are also claims that the maintenance (delay) period is dissociable by cognitive load when FC is considered (as opposed to activation-only). While the statistical testing to support this is well-documented in the supplement, it appears that the overall pattern driven by 'load' (and discernable via FC) involves a divergence during encoding and subsequent convergence during maintenance (e.g., fig 3cii and 3ciii). These patterns should be assessed (e.g., compare the nonlinear functions of each 'load').

5. The study appears to control for simple sensory-motor factors when classifying many of the cognitive factors of interest. This is critical for isolating truly cognitive (as opposed to perceptual or motor) effects. However, it appears that the "stages" classifications were not well controlled in this respect. Instead, it appeared that basic visual and motor differences differentiated these stages (in contrast to the WM domain classifications). It would be important to acknowledge this major limitation of the findings, and consider ways to limit its impact either in these analyses or in future studies.

6. Pg. 16: It appears that cluster-based thresholding was used to correct for multiple comparisons for the conjunction analysis. The issues demonstrated by this paper would need to be taken into account for valid statistical inference: Eklund A, Nichols TE, Knutsson H (2016) "Cluster failure: Why fMRI inferences for spatial extent have inflated false-positive rates". *Proc Natl Acad Sci USA.* 201602413-201602416. <http://doi.org/10.1073/pnas.1602413113>

7. Pg. 16: It is unclear how multiple comparisons were corrected for the initial GLM estimates.

Were the GLM estimates corrected for multiple comparisons? How?

8. Several of the analyses appeared to cut corners, with the complexity of the analyses making it difficult for the reader to find these cut corners. For instance, Figure 5C is highly abstract and based on a complex (potentially unnecessarily complex) analysis, making it difficult to interpret. In the text it is stated that "the domain specific configurations that emerge are clearly augmented as a function of stimulus load", yet this is not clear from the figure. Further, even if this were clear from the figure it must be shown to be the case statistically in order to make it a scientifically valid claim. There appear to be a number of cases like this throughout the manuscript, where careful statistical analyses are needed to back up claims, and the authors should consider simplifying their analyses (because they are unnecessarily complex, limiting their interpretability). As a brief other example, Figure 2B is very complex, and unnecessarily so.

Minor Considerations

1. The domains of WM included in this study are characterized by which aspect of stimulus presentation is cued for a later discrimination. These include: number tags on the screen, fractal images next to those numbers, and spatial position of those sets. The text initially describes these as visual domains; however, it is not clear that they are all readily characterized as 'visual'. Spatial position may refer to early visual processing (e.g., retinotopic encoding of the visual field in V1) or associational representations of objects in space, or both; fractal representations may relate to associational object recognition; and numerical representations might additionally recruit processing outside any putative visual network (e.g., above and beyond multimodal recruitment instantiated via position and fractal design), such as those biased to process semantics.

Importantly, the number domain is later described as "verbal WM" (page 8), which confuses the prior categorization of this as a visual domain of processing. While the purpose of this study is to move beyond traditional localist views, it is still important that task-based manipulations are discussed in a principled manner (e.g., operationalized when possible). Moreover, the WM networks invoked by this paradigm do not discount the possibility of a completely different set of findings for a task that is based in audition. Therefore, consideration should be made to discuss this as visual-semantic-WM network coding, as opposed to generalizing across WM.

2. Another consideration pertains to the analysis that challenges the view that regions of the lateral frontal lobe preferentially relate to maintenance and manipulation during WM (page 7). In the spirit of considering whole-brain-derived representations (e.g., unbiased by conjunction for a different paradigm dimension, such as in DG and SG), it would be particularly useful to know what voxels (or vertices, depending on the implementation) were included in the 'WB' set. Moreover, could these analyses be constrained to LPFC regions (via seeding), given that there is an a priori hypothesis in this part of the study?

3. Pg 1: "Extending a recent exploratory analysis of many functional anatomical mappings[24] in this journal...". It appears that reference 24 is actually only a preprint. Also, it is unclear how the present study is an extension of reference 24.

4. The study involved only 19 subjects (study 1) and 17 subjects (study 2). A standard power analysis in G*Power [Faul F, Erdfelder E, Buchner A, Lang A-G (2009) "Statistical power analyses using G*Power 3.1: Tests for correlation and regression analyses". Behav Res. 41:1149-1160.<http://doi.org/10.3758/BRM.41.4.1149>] indicates that you would need a relatively large effect size to detect effects with this sample size at 80% power ($d > 0.67$). Given that this is generally considered to be a large effect size, the issue of likely false negatives given the small sample size should be addressed in the Discussion.

5. Pg. 15: The model of the 3T Siemens scanner should be specified in the Methods.

6. Pg. 15: "implantation"; should be "implementation"?

7. Pg. 15: "A multivariate outlier's detection analysis was performed to detect low SNR values." This should be better specified; it is unclear what was done.

References

Baddeley, A. (2000). The episodic buffer: a new component of working memory? Trends in

Cognitive Sciences,
4(11), 417-423. [https://doi.org/10.1016/S1364-6613\(00\)01538-2](https://doi.org/10.1016/S1364-6613(00)01538-2)
Fletcher, P. C., & Henson, R. N. (2001). Frontal lobes and human memory: insights from functional neuroimaging.
Brain: A Journal of Neurology, 124(Pt 5), 849-881.
Glasser, M. F., Coalson, T. S., Robinson, E. C., Hacker, C. D., Harwell, J., Yacoub, E., ... Van Essen, D. C. (2016). A multi-modal parcellation of human cerebral cortex. *Nature*, 536, 171.
Owen, A. M., Evans, A. C., & Petrides, M. (1996). Evidence for a two-stage model of spatial working memory processing within the lateral frontal cortex: a positron emission tomography study. *Cerebral Cortex* (New York, N.Y.: 1991), 6(1), 31-38.
Roy, A. (2017). The Theory of Localist Representation and of a Purely Abstract Cognitive System: The Evidence from Cortical Columns, Category Cells, and Multisensory Neurons. *Frontiers in Psychology*, 8.

Response to Reviewer #1

“Dynamic network coding of working-memory domains and working-memory processes,” Soreq, Leech, and Hampshire. This manuscript describes the results from several multivariate and connectivity-related analyses of fMRI data from two sets of working memory tasks, different subjects performing each, with results from one experiment validated against the results from the other. The idea is to identify and characterize high-dimensional brain states that underlie different WM processes (encoding, maintenance/manipulation, retrieval) crossed with different stimulus domains (spatial, object, digit). This is an important undertaking, and one that, a priori, should yield valuable information about large-scale dynamics of the neural bases of WM function. I had difficulty, however, with what reads like a recitation of ad hoc interpretations of results that, albeit generated by sophisticated methods, don’t leave me with a better understanding of WM than what I brought to the manuscript. Additionally, the authors report several puzzling failures to replicate well-established findings in the field. It should be the case that, unless what the authors characterize as “early neuropsychological perspectives” are outright wrong (and this would mean that, in some cases, results from tens or more different groups around the world have been wrong), the dynamical network-level methods brought to bear on this dataset could show that the “early”, often univariate, results hold, but perhaps their interpretation changes, or they can be shown to be superficial by products of more fundamental processes that are captured by these more sophisticated analyses.

General response to Reviewer #1

We thank the reviewer for taking the time to review our manuscript and for recognising the importance of the endeavour along with the sophistication of our analysis approach. We have modified our manuscript to emphasise the vital novel insights more clearly and informatively. We also include additional analyses of a more ‘classic’ type in our supplemental materials and add a new summery figure that attempts to distil the key findings and synthesise them with previously reported findings.

We note that the only major finding from the classic literature that we were not able to replicate was the manipulation construct effect. Our dynamic network analysis accounts for all other effects that we would expect based on the literature. As per the reviewer’s suggestion, in the revised manuscript we have tried to emphasize the point that our findings support the synthesis of several classical standpoints from both behavioural and imaging literature while extending them and incorporating them within a more modern multivariate/network science perspective.

As noted by reviewer #2: *“This study highlights the importance of network science and machine learning in building theory about complex systems.”*. We believe that the strength of our approach is the way we incorporate modern data-driven methods to produce evidence in an

unbiased manner. Moreover, as reviewer #1 suggests, we show classic univariate analysis results to be by-products of more complex multivariate network processes. Furthermore, most of our predictions, which are based on and extend classic theorising, were validated and replicated in a separate dataset. The sole exception to this is the lack of a 'manipulation' effect. We discuss this in more depth in the revised manuscript, but note, that this is a single surprising instance, not several as suggested above.

To address these points and other raised by the reviewer, we took the following actions:

1. *Our updated manuscript emphasises the following points more explicitly:*
 - a. *We have reworked the paper to make clear that we are providing a multivariate network perspective that does indeed explain, and thereby allow us to synthesise, many of the standpoints from the classic WM literature.*
 - b. *We refer to the specific prediction regarding expected load effects in the introduction*
 - c. *We amended the results section to make clear that effects of activation load are primarily evident at the network connectivity level.*
 - d. *We also added a line in the discussion addressing this issue*
 - e. *We extended our explanation concerning our inability to find any evidence of a manipulation effect*
 - f. *We reworked our introduction to better reflect the tension between classical and current WM perspectives*
 - g. *We fixed typos in both text and references*
2. *We added to the supplementary material the following new analyses for completeness*
 - a. *We added a per ROI univariate analysis examining load effect during maintenance (Supplementary table 54)*
 - b. *We added targeted DLPFC ROI linear mixed effects analysis to examine as sensitively as possible univariate manipulation effects (Supplementary table 53)*

We hope that the reviewer will agree that the additional emphasis that we place on integrating our various findings while extending them to the network domain makes the paper more interpretable regarding the field at large. However, we are very open to further suggestions regarding where such emphasis might be extended. We thank the reviewer as we feel that our manuscript is greatly improved.

Specific responses to Reviewer #1

1. Absent this ... **What IS persuasive, in contrast, is if one can show the classical pattern, but then apply the fancy new ideas and analyses to argues that this activity is better understood ..."**

We agree entirely with the approach as outlined by the reviewer; only one of the predictions expected based on the classic-mass univariate approach of the classic literature could not be accounted for in our analyses. We have made this clearer in the revised manuscript.

2. "In accordance with our prediction, within the DG ROIs, there was no significant increase in activation during maintenance at higher load." ... **there is no place in the Introduction, or elsewhere, in which this prediction had been articulated...**

This is a crucial point, in fact the prediction based on network science theory was that connectivity should provide a stronger indicator of load than regional activity – however, this was trimmed from the previous version of the introduction. We have updated the main text accordingly.

3. The second concern is the failure to see a load effect...

We feel that perhaps our focus on the global effect was misinterpreted as failure to detect specific ROI effects as a function of load. To address that point and to allow comparison to the classic localist literature, we have now conducted a more detailed univariate inspection of ROIs that are affected as a function of load during **maintenance** and added them to the supplementary material. As one would suspect given the conjunction plots, individual ROI that are significantly more active at higher loads are the parietal, precentral and insula. This finding provides a further replication of previous work (Ikkai, McCollough, & Vogel, 2010) and has been added to the results and discussion in the revised manuscript.

4. Furthermore, some more recent studies that have applied multivariate analysis methods to data with load manipulations have reported that, as is the case with behavior, multivariate indices of the neural precision of stimulus information in WM declines monotonically with load (studies from Awh, Postle, Serences spring to mind). This is the opposite of what is reported here.)

We assume that the reviewer is referring to (Emrich, Riggall, LaRocque, & Postle, 2013). They report "load-dependent decrease in classification sensitivity during the delay period", which on the surface is in contrast to our findings. It is important to note though that there are many differences between these two studies, in fact they differed in what was measured and what was manipulated so are hard to compare at all. More specifically, Emrich et al., used a voxel-wise as opposed to connectivity-based classification. Second, load is defined as the proportion of coherent directional motion (i.e. less relevant signal = higher load), as opposed to number of elements to remember. Both studies manipulate difficulty, but one could argue that the coherence manipulation affects perception and encoding as opposed to maintenance load. We

also note that we replicated the load-connectivity effects across two separate datasets, so we have high confidence in the validity of the results. We now reference this paper in the main article.

5. The same problem is true for maintenance vs. manipulation ...

The reviewer raises one of the most surprising aspects of the study. No one was more surprised than us when this contrast showed no differences. In fact, the senior author worked for many years in the lab of the researcher who proposed the two-stage model of WM, so we had a strong prior belief that this would work. A key point to make is that this is, in fact, the only comparison from the classic literature that we failed to observe despite trying a range of analysis approaches (e.g. classic univariate voxel wise analyses conducted with liberal uncorrected thresholds, ROI analyses uncorrected, and of course our reported classification analysis.)

The question is why we saw no such effect? Several possible explanations exist for this disconnect between our results and those of some previous studies (although we do note, not everyone has subscribed to the 'two-stage' model of working memory).

- The data could have been poorly analysed. We believe this is unlikely given the high reliability for all other contrasts, including when matched for number of features and number of observations.
- The manipulation could have been too easy/fast to detect. This is conceivable, but we do not believe it to be the case because people reported finding this challenging. Certainly, from a phenomenological perspective, it takes most of the 10 second timeframe to apply the manipulation.
- Our search might have been too spatially broad, inflating false negatives. However, as suggested by Reviewer #2 we also identified an ROI that correspond to Cowen et. al. publication and tested local effects using linear mixed effects model. No such effects were found (see supplementary material) and nor were they evident voxelwise and uncorrected.
- Another possibility is that due to small sample (N=16) we have the effect is undetected due low power; however, this seems equally unlikely considering the large effect reported by Cowen et. al. (i.e. $F=5.08$) that suggests that for five selected ROI a sample of fewer than 13 subjects is sufficient.

Our view is that the failure to observe this effect may relate to the design of the task, which uses a retro cue. Possibly it is the maintenance of the manipulation rule rather than its application that produces heightened activity within the DLPFC. This would explain the fact that we have previously observed that strong DLPFC activation increases when solving problems that involve more rules during reasoning tasks (as reported in several of our previous articles). Resolving this issue requires a further focused study that we feel is outside of the scope of the current paper

which, we note, has many other significant findings. In the modified article, we have reworked our discussion of this point.

6. Some of the broad generalizations don't accurately capture the current state of the field. For example, the very first sentence of the introduction is a statement that would have worked in the 1990s, but that an increasingly smaller number of scientists have subscribed to since that time. (Indeed, one of the two reviews cited argues for an explicitly contradicting position.)

Following both your comment and Reviewer #2 comments, we made specific changes to the introduction that note the evolution of the field in the past two decades.

7. Another example is that the multiple demand (MD) system "would be predicted to show the least difference between [WM domain and WM process]. This is a mischaracterization, in that the point of the MD system is its ability to flexibly configure itself to accommodate whatever challenges an individual encounters, not unlike the "flexible coding" properties of pyramidal cells of the PFC

We specifically selected areas of the brain that are active for all domains and processes. These should be the hardest areas for classifying specific domains and processes as they show the greatest similarity across them. Yet if the system is highly flexible, as proposed by some (e.g., Duncan/Cole etc), then a question arises as to whether we are looking at a uniform system that supports a highly generalisable process or a heterogeneous system that can adapt into various configurations to support different processes. In the latter case, we should see strong coding of task dimensions even just within this volume, i.e., ignoring its connection to the external system. Whereas from the former, it should appear homogenous internally at the network level. This cuts to the heart of one of the main questions we were addressing, which in turn informs our understanding of how MD codes for different tasks, i.e., via local neuronal representations or also global connectivity patterns? This is a fundamental question and we believe our results clearly favour the latter case. Furthermore, we believe it to be highly informative and novel that domain by process interactions are only decodable by connectivity states within MD, which again indicates a network-coding mechanism.

8. The insula is not a subcortical structure!

Thank you for pointing this out, we have now corrected it.

9. Throughout the manuscript there is an overreliance on abbreviations that are nonstandard, idiosyncratic to this study, making it unnecessarily difficult to read (e.g., "WB" "SG" "DG" "FTP" "IPT"...)

This is a good point, we have edited heavily to remove abbreviations with a few key exceptions. We also provide a table defining the abbreviations for reference in the supplementary materials.

10. The text refers to "Extending a recent exploratory analysis of many functional anatomical mappings[24] in this journal;" but ref 24 is, in fact, a bioRxiv preprint.

Ref 24 was recently accepted to Nature Communications, as two of the three authors are also senior authors on that paper we were privy to that information in the time of submission. We have updated the article to reference the published article.

11. "implantation"

Thanks – this has been corrected

12. Refs 3 and 5 have errors.

Thanks – this has been corrected.

Response to Reviewer #2

Summary & Merits

The primary claim in this study is that localist frameworks for describing working memory (WM) (Baddeley, 2000; Fletcher & Henson, 2001; Owen et al., 1996; Roy, 2017) are insufficient, and in turn, a multivariate 'network coding mechanism' is more apt. Using data-driven dimensionality reduction methods to identify and assess WM networks by domain of processing, stage of processing, and demand on processing, followed by context-dependent psychophysiological interaction (gPPI), and multi-class pattern analysis, the results provide convergent support in this direction. ***It is noted that localist findings can be reconciled as an overly-sparse rendition of the network-space instantiated in WM.*** This allows for unification between studies (and generations of research with increasingly advanced acquisition and analytic techniques), as opposed to invalidating previous findings. This study highlights the importance of network science and machine learning in building theory about complex systems.

General response to Reviewer #2

We thank the reviewer for their positive and helpful comments. The clear way in which you summarised our approach and findings is more than our own from the original article. We have modified our manuscript to address both major and minor points raised. This has been a substantial job requiring additional analyses and a new framework figure to convey the logical steps taken and how they advance our understanding of working memory.

We took the following actions to address your comments:

1. The introduction requires further development. Specifically, *the localist framework* that this study is refining via network solutions should be delineated more thoroughly to give proper context to the findings.

We have added specific changes to the introduction to summarise succinctly what we perceive to be the key difference between the localist and network perspectives, including supporting citations.

We define the localist framework as the ongoing attempt to map aspects of cognition, e.g., the behaviourally distinct information domains and the process of WM, to functionally specialised regions of the brain. This classic approach seeks to map brain mechanisms that underlie cognitive tasks using circuit diagram where each anatomical component is ascribed a specific operation in a one-to-one manner, e.g., encoding spatial information and manipulating such information have been proposed to map to different regions of the lateral frontal cortex. Implicit in this is the assumption that greater processing demand of one or other type should selectively increase activity within the discrete neuroanatomical module that is responsible for that process.

In contrast, the 'network coding perspective' suggests that the relationship between cognitive processes and brain activation is many-to-many as opposed to one-to-one. Specifically, task-relevant information exists across transient coalitions of many brain regions, and goal-directed processing of that information recruits a complex web of connections within this network, i.e., as opposed to dedicated pathways of an inflexible circuit. A prediction of this is that the behaviourally distinct information domains and sub-processes of WM should be most reliably evident as multivariate patterns of connectivity, i.e., 'transient states', potentially within a network of brain regions that are engaged in all of those processes.

2. Moreover, there is mention of a 'unifying framework' (page 2) that is not explicated or synthesized in the text. *While one may discern that this framework is the 'network coding mechanism' mentioned throughout, it should be explicitly labeled as such.*

We have updated the text to explicitly label this as recommended.

3. There are various points throughout the analytical pipeline where alternate methods could be used. While the methods section and supplement do disclose the rationale behind many of these choices, such justifications should be recapitulated in the main text as well (salient examples: convolution method, functional connectivity estimation, cross-study dice coefficients, match/mismatch accuracies, and null-model generation). Moreover, there were instances when a methodological choice was either not adequately justified, or was novel and so required more interpretational detail.

There is now additional brief text to overview the rationale in the main text followed by a more detailed supplementary explanation. There is also a link to a toolbox that contains all the analysis pipeline tools along, also in supplementary material.

4. Firstly, the use of data-driven conjunctions and parcellation schemes to derive the WM clusters that are either 'domain-general' (DG) or 'state-general' (SG) is not well justified against the alternative of a whole-brain, multi-modal approach (such as Glasser et al., 2016).

As a group, we have an interest in the multiple demand cortex (which the two data-driven parcels sets try to capture) – it has been the senior authors primary focus in much of his past work due to the broad role that this volume of the brain plays in cognition. A crucial point in our study is to contrast between the amount of working memory relevant information contained within the two data-driven subsets. As a result, we use the Finn whole brain atlas as a baseline measure capturing all information available. Importantly, this is the reason why using the cortical only Glasser atlas is problematic as a valid comparison between surface based ROI (Glasser), and volume based ROIs (DG and SG) is not trivial. The fact that domain classification models trained

from information within the DG set are as accurate as the much more complex whole brain models reflect that even though the DG is active for all domains the multivariate pattern within it is extremely sensitive to the different domains, which we consider to be a fundamental insight into how MDC operates.

5. Additionally, the use of a pre-defined ROI set can help avoid circularity in analyses (see [Kriegeskorte N, Simmons WK, Bellgowan PSF, Baker CI (2009) "Circular analysis in systems neuroscience: the dangers of double dipping". Nat Neurosci. 12:535-540. PMID: PMC2841687 <http://doi.org/10.1038/nn.2303>].).

Indeed, double dipping is a major concern that we were very aware of throughout the analyses; to address this we took several precautions and this was the main reason for including the pre-defined whole brain ROI set. Importantly, while we stress the fact that conjunctions from both studies look the same, our parcellation sets are defined based on conjunctions from study 1 only. Furthermore, while we believe that within study performance is informative, our reported statistics are based on applying models trained on study 1 and tested on study 2 (a completely independent study that is similar but not identical to study 1). This completely eschews the problem of double dipping.

6. A predefined whole-brain ROI-set is used for a portion of the analyses, but it is not analyzed with the same level of detail as the DG or SG set. Additionally, the DG and SG sets suggest an intersection or collapse amongst domains and stages, respectively, however, they are later used in supervised learning algorithms that classify upon domain and stage (varied depending on the specific analysis). This is difficult to interpret, and requires more thorough consideration for the network coding mechanism.

We have now extended our report of whole brain analyses. Due to space constraints and our particular interest in MDC, some of these analyses are reported in the supplementary materials.

7. Alternatively, a whole-brain parcellation could be applied to the GLM data stacks (e.g., what is described in the "Exploratory analyses" portion of the methods section), and community structure (via HCL or maximized modularity) could be derived, naïve of task condition (e.g., what is presumably utilized as the WB ROI set).

We have now included analyses that include the whole brain summarised according to an established framework of resting state connectivity networks. These analyses are now included in the main text as well as in figure 5. We have also added a new unsupervised analysis highlighting the fact that the natural clustering of events revolves around the processing stages even when examining activity from the entire brain and at the mini block level (see figure 2).

8. Secondly, the term 'rest' is used in interpreting the results of figure 3, but it is not clear how this is defined. The text suggests that the rest period is when the mean activity in visual clusters has a beta-weight of zero, which occurs during maintenance (also called 'delay' in some figures). This is then used to compare against frontoparietal and visual

regions, for both activation and connectivity during encoding and delay. In the case of contrasting visual-to-visual, it is thus implied that comparing delay to rest is a self-comparison, although this is not clear.

We agree that the term rest is somewhat ambiguous. To clarify, in rest we meant the inter-trial interval, i.e., the 10s period between individual trials, where no explicit action is required from the participant. This acts as our implicit baseline and the whole model is relative to these periods. To make this completely clear we have changed rest to inter-trial-interval in both text and figures and added a general panel in figure 1 that clarifies the stages of the task.

9. In the case of contrasting frontoparietal-to-visual, this raises the question of why a visual-system-determined baseline would be used (as opposed to a FPT baseline)?

In this simple analysis, we compare encoding to maintenance for all functional clusters individually, using multiple paired t-test. The only baseline here is the implicit baseline for the relevant area, which includes brain activity during the 10s ITI.

10. Regarding the same figure (3c), there are also claims that the maintenance (delay) period is dissociable by cognitive load when FC is considered (as opposed to activation-only). While the statistical testing to support this is well-documented in the supplement, it appears that the overall pattern driven by 'load' (and discernable via FC) involves a divergence during encoding and subsequent convergence during maintenance (e.g., fig 3cii and 3ciii). These patterns should be assessed (e.g., compare the nonlinear functions of each 'load').

While this convergence and divergence pattern is indeed fascinating, we felt that the visualization was misleading as it gave the impression of a temporal continuum. We now present histograms that better represent the different underlying distributions and highlight the differences between the processing stages and how increased load affects them. We feel that these differences are now easier to understand and better reflect the actual data than our previous figure.

11. The study appears to control for simple sensory-motor factors when classifying many of the cognitive factors of interest. This is critical for isolating truly cognitive (as opposed to perceptual or motor) effects. However, it appears that the "stages" classifications were not well controlled in this respect. Instead, it appeared that basic visual and motor differences *differentiated these stages* (in contrast to the WM domain classifications). It would be important to acknowledge this major limitation of the findings, and consider ways to limit its impact either in these analyses or in future studies.

We have further emphasised this point in the main text. Specifically, we stress the point that basic perceptual and motor demands of the task drive the natural clusters of the data (see the new whole brain tSNE clustering, as well as the hierarchical cluster analysis), which is as expected.

However, the different domains may nonetheless be classified with similar accuracy based on the multivariate analysis.

12. Pg. 16: It appears that cluster-based thresholding was used to correct for multiple comparisons for the conjunction analysis. The issues demonstrated by this paper would need to be taken into account for valid statistical inference: Eklund A, Nichols TE, Knutsson H (2016) "Cluster failure: Why fMRI inferences for spatial extent have inflated false-positive rates". Proc Natl Acad Sci USA. 201602413-201602416.<http://doi.org/10.1073/pnas.1602413113>

We have added a line in the text regarding our choice of cluster correction threshold. Notably though, as the subsequent steps move to ROI analysis, if there were any false-positive inflation this would increase the noise in the data prior to classification. The fact that we find strong and reproducible effects across studies means we have high confidence that the results do not reflect false positives.

13. Pg. 16: It is unclear how multiple comparisons were corrected for the initial GLM estimates. Were the GLM estimates corrected for multiple comparisons? How?

We have now clarified how multiple comparisons were performed for each section in the online methods.

14. Several of the analyses appeared to cut corners, with the complexity of the analyses making it difficult for the reader to find these cut corners. For instance, Figure 5C is highly abstract and based on a complex (potentially unnecessarily complex) analysis, making it difficult to interpret. Further, even if this were clear from the figure it must be shown to be the case statistically in order to make it a scientifically valid claim.

We are grateful to the reviewer for this comment. Exploring and explaining complex multivariate patterns is a challenge. In this paper, we used and developed an array of visualization approaches which the first author developed and implemented (the code is shared as a visualisation toolbox via GitHub). In this case we agree that we took a sub-optimal approach. To amend this, we completely redesigned figure 5 adding two new sections, that replace the former section, these panels first show the trial global and specific increase in connectivity as a function of load, and then move to a large-scale, per connection, general linear modelling approach that statistically examines the effect of domain, load and their interaction (when controlling for study).

Interestingly, we show that domain effects during maintenance relate almost entirely to the inter and intra frontoparietal cluster connections, which reinforces the view that these frontoparietal areas are central to the maintenance process. This analysis has been added to the paper following this helpful comment.

15. There appear to be a number of cases like this throughout the manuscript, where careful statistical analyses are needed to back up claims, and the authors should consider

simplifying their analyses (because they are unnecessarily complex, limiting their interpretability). As a brief other example, Figure 2B is very complex, and unnecessarily so.

While we agree that figure 2B is complex, in the spirit of transparency and representing the data as-is we feel it is the best we could achieve. The main idea of Figure 2B is to first show that the natural clustering of the activation data revolves around the simple perceptual and motor processes (i.e. stage clustering Fig 2Bi), followed by purity analysis to quantify how strong this effect is, and finally projecting the functional clusters back onto the data-driven ROI uncovered by the watershed algorithm.

16. or schematic that synthesizes the findings would be beneficial for a multi-dimensional study such as this (e.g., a visualization of what is touched upon in the discussion section).

We thank you for this suggestion. We have added Fig 6 and hope it addresses this recommendation adequately.

Minor Considerations

1. The domains of WM included in this study are characterized by which aspect of stimulus presentation is cued for a later discrimination. These include: number tags on the screen, fractal images next to those numbers, and spatial position of those sets. The text initially describes these as visual domains; however, it is not clear that they are all readily characterized as 'visual'. Spatial position may refer to early visual processing (e.g., retinotopic encoding of the visual field in V1) or associational representations of objects in space, or both; fractal representations may relate to associational object recognition; and numerical representations might additionally recruit processing outside any putative visual network (e.g., above and beyond multimodal recruitment instantiated via position and fractal design), such as those biased to process semantics. Importantly, the number domain is later described as "verbal WM" (page 8), which confuses the prior categorization of this as a visual domain of processing. While the purpose of this study is to move beyond traditional localist views, it is still important that task-based manipulations are discussed in a principled manner (e.g., operationalized when possible). Moreover, the WM networks invoked by this paradigm do not discount the possibility of a completely different set of findings for a task that is based in audition. Therefore, consideration should be made to discuss this as visual-semantic-WM network coding, as opposed to generalizing across WM.

We agree and we now use visual-based labels. I.e., spatial, fractal-pattern and number. Importantly, the former two are considered in classic imaging to tap different visual processing streams. Defining the latter is a more complex issue as it relies on verbal processing areas that

are supposedly rearticulated to support efficient maintenance. Therefore, whilst presentation is visual, encoding and maintenance is probably activating both visual and verbal processing streams.

2. Another consideration pertains to the analysis that challenges the view that regions of the lateral frontal lobe preferentially relate to maintenance and manipulation during WM (page 7). In the spirit of considering whole-brain-derived representations (e.g., unbiased by conjunction for a different paradigm dimension, such as in DG and SG), it would be particularly useful to know what voxels (or vertices, depending on the implementation) were included in the 'WB' set. Moreover, could these analyses be constrained to LPFC regions (via seeding), given that there is an a priori hypothesis in this part of the study?

We have now added to the supplementary material a WB ROI table with central voxel locations. We also examined five ROI that are close to the previously reported manipulation areas (see the new supplementary table 54) with no effects found. Notably, no effects were found even when performing uncorrected contrast manipulation > non-manipulated.

3. Pg 1: "Extending a recent exploratory analysis of many functional anatomical mappings[24] in this journal...". It appears that reference 24 is actually only a preprint. Also, it is unclear how the present study is an extension of reference 24.

The link between the two papers is that the very general conclusion's are quite similar: specifically, it is not correct to map individual tasks or cognitive operations to individual brain regions or circuits in an exclusive one-to-one or one-to-many manner. Instead, there appears to be a many-to-many, i.e., multivariate coding mechanism. This holds even for the most commonly activated brain regions that constitute multiple demand cortex. This cuts to the heart of what we consider the network science perspective to be. We have tried to clarify this point in the text.

4. The study involved only 19 subjects (study 1) and 17 subjects (study 2). A standard power analysis in G*Power [Faul F, Erdfelder E, Buchner A, Lang A-G (2009) "Statistical power analyses using G*Power 3.1: Tests for correlation and regression analyses". Behav Res. 41:1149-1160.<http://doi.org/10.3758/BRM.41.4.1149>] indicates that you would need a relatively large effect size to detect effects with this sample size at 80% power ($d > 0.67$). Given that this is generally considered to be a large effect size, the issue of likely false negatives given the small sample size should be addressed in the Discussion.

Calculating power for multivariate machine learning analyses of this type is non-trivial compared to univariate correlation and cross-condition analyses. Likely it would require the running of complex permutation analyses, which given the multivariate nature of the data, would in turn require many assumptions regarding the strengths, distributions and collinearities of effects within

that multivariate space. For the majority of our analyses, we confirm results using what is essentially a train-test approach across independent datasets and using multiple features. Furthermore, we use a stack of observations from each individual as opposed to a single measure per individual. Therefore, the robust and reproducible findings that we report highlight the strengths of our multivariate machine learning analysis approach. We now specifically make this point in the main text.

- Emrich, S. M., Riggall, A. C., LaRocque, J. J., & Postle, B. R. (2013). Distributed patterns of activity in sensory cortex reflect the precision of multiple items maintained in visual short-term memory. *Journal of Neuroscience*, *33*(15), 6516-6523.
- Ikkai, A., McCollough, A. W., & Vogel, E. K. (2010). Contralateral delay activity provides a neural measure of the number of representations in visual working memory. *J Neurophysiol*, *103*(4), 1963-1968. doi:10.1152/jn.00978.2009

Reviewers' comments:

Reviewer #1 (Remarks to the Author):

"Dynamic network coding of working-memory domains and working-memory processes," Soreq, Leech, and Hampshire. Although the authors have responded to all the concerns raised in the first round of reviews, I still feel ambivalent about this paper. The authors acknowledge the importance of, and have made some steps toward, integrating their results with the "classical," "localist" literature, and this is appreciated, but there's still a part of me that's not sure that I understand the neural bases of working memory any better now than I did before reading this paper. Admittedly, part of this is due to the complexity and sophistication of the analytic methods, but for a general-readership journal like Nature Communications, one expects that the main point of the paper will be accessible to neuroscientists coming from a broad range of fields, and I just don't think that this manuscript achieves this. Much of the writing comes across as so much "word salad," epitomized by the final sentence of the manuscript. In other places, there seem to be outright contradictions. For example, there's the conclusion from Figure 3d that "domain-specific patterns of connectivity were enhanced as a function of WM load," which is difficult to reconcile with the conclusions arising from Fig. 5 bi and bii. Looking at these figures the point seems to be that there are distinct connectivity patterns for Spatial vs. Fractal vs. Digit, but then a more-or-less common pattern of upregulation of connectivity between MD regions and visual regions is observed when load is increased, regardless of domain? It's asserted that the latter pattern argues against 'mutual exclusivity' of discrete representation of domains. This would seem, on the face of it, to argue against the claims arising from Figure 3. And while I'm thinking of Fig. 5 bi and bii, isn't an equally plausible description of these findings that domain stimulus domain is represented via very distinct "multiplex topological shapes [of connectivity patterns]" and then that, orthogonal to this, a different state that is common across domains waxes and wanes as the task gets more or less difficult?

The Discussion asserts that one of the results "explains some of the confusion within the classic literature," but then it's not clear what this purported confusion is, nor are any citations offered.

Another major stumbling block for me continues to be the failure to see a univariate maintenance vs. manipulation effect in Study 2. This is the kind of effect that's seen at the single-subject level, that can be reliably used, for example, as an n-of-one study in an undergraduate neuroimaging class that's more-or-less guaranteed to produce a result. The retrocuing account that the authors propose doesn't seem plausible, because it would mean that subjects effectively ignore the retrocued and still carry out the manipulation on all trials. If that were true, one wouldn't expect a behavioral difference between the two trial types. I worry that this 'quirk' in the data might call into question, for some readers, the fundamental quality of the data being analyzed. (Somewhat relatedly, there are important descriptive statistics missing: what were accuracies and RTs broken out by condition?)

Introduction: In the paragraph beginning "Second" I believe that the citations are flipped?

Reviewer #2 (Remarks to the Author):

Overall, the revised manuscript represents a significant improvement over the original version. The logic of the study is more clearly delineated, terms are more consistently presented, and the introduction/discussion sections are synthesized in a more easily comprehended manner. Furthermore, the authors did a lot of work to address specific concerns raised by all reviewers, including the new schematic (Fig. 6) that summarizes important results and distills the findings

into a more cohesive narrative. That being said, select analyses, results, and visualizations are still fairly dense and difficult to parse out at times. However, the take away messages of the study are clearer now, and while some readers may find it difficult to follow the detailed aspects of the results, most readers should be able to comprehend the network-coding mechanism(s) being demonstrated. Moreover, while there are some concerns about “accessibility” of the findings, the questions addressed in this study along with the rigor of analyses are important for the field. Listed below are major and minor concerns moving forward; major concerns refer to resolving the “denseness” of the results section, and minor concerns are more technical in nature (i.e., more easily addressed).

Major concerns

1. Page 6: what is the logic behind switching focus to frontoparietal and visual ROIs? This might be clear to some readers, but 1 sentence explaining this would be helpful.
2. Pages 6 – 7 and Figures 3 & 4: the details of these results were difficult to follow; more text on the maintenance-related findings would be helpful. For example, there was lowered global activity during maintenance relative to other stages, but connectivity analyses show more stability. However, there were influences/interactions based on which regions were in focus (visual and/or frontoparietal), cognitive load, inter/intra connectivity, and exactly which stages were being compared. Overall, the complexity of this was hard to follow such that there was not a clear take-away from the results/figures. This was compounded by the classification analyses, which additionally raised the issue of how domain-general and stage-general patterns related to each other. A simple way to remedy this might be to connect the main-text more explicitly with the figures. This would ultimately add more text to the main body of the results section, but would help comprehension. For example, the distributions in Figure 3b-c could be explicated in the text as well (this suggestion could be applied throughout these sections – focusing on figures 3 & 4).

Minor concerns

1. General: consider having someone triple check that the panel letters for figures match what is in the text and that all panel letters are consistently written in figure captions
2. General: probe/recall/retrieve seem to be used as synonyms; consider choosing 1 term for consistency
3. Page 2, line 20: the term phonological is used here but nowhere else; consider replacing it with ‘number’ to be more consistent with what the study is directly manipulating
4. Page 4, line 9: “fractal patterns” should be singular for consistency
5. Page 5, line 19: “study 2” in this syntax seems to be missing
6. Page 5, lines 14-21: consider using the past tense exclusively
7. Page 17, line 24: capitalize “procedure”
8. Methods section general: use consistent tense
9. Page 28, Figure 3a: consider adding a scale for the connectivity matrices
10. Page 28, Figure 3c: consider stating which regions were used in the averages here
11. Page 29, Figure 4: the text is cut off in the pdf
12. Page 29, Figure 4a ii-iii: consider writing what colors indicate
13. Page 31, Figure 6: typo in ‘parcellation sets’ panel; fix to “unbiased”
14. Page 31, Figure 6: random words seem to be capitalized in the text descriptions

Response to Reviewer 1

1) A major stumbling block for me continues to be the failure to see a univariate maintenance vs. manipulation effect in Study 2. This is the kind of effect that's seen at the single-subject level, that can be reliably used, for example, as an n-of-one study in an undergraduate neuroimaging class that's more-or-less guaranteed to produce a result. The retrocuing account that the authors propose doesn't seem plausible, because it would mean that subjects effectively ignore the retrocue and still carry out the manipulation on all trials. If that were true, one wouldn't expect a behavioral difference between the two trial types. I worry that this 'quirk' in the data might call into question, for some readers, the fundamental quality of the data being analyzed.

As communicated in our previous response, we did share the reviewers surprise at the lack a manipulation effect in the convolved model. To address this issue more satisfactorily, we have conducted a supplementary analyses focused on the activation timecourses from four frontoparietal regions of interest that have been associated with WM manipulation in the literature. These ROIs were defined at the peak activation coordinates as reported in the classic article of *Veltman, Rombouts and Dolan 2002*, in which they contrasted WM tasks that had explicit manipulation vs. maintenance demands. We first examined the levels of activation at the end of the maintenance and manipulation blocks - that is, just prior to the retro-cue when we would expect no difference, and then just prior to the probe when we would expect any such effect to be approaching its peak. We are pleased to report that this analysis showed a statistically robust increase in activation at the group level for the manipulation vs. the maintenance contrast in all four ROIs just prior to the probe. As expected, no such differences were evident prior to the manipulation retro-cue.

Critically though, when we plotted the entire averaged time course through the trials for these ROIs, i.e., anchored to the end of the rest period, it was clear that they were also sensitive to WM load and stimulus domain. This lack of a specific network pattern associated with maintenance explains the failure of the classifier, which operates on multivariate patterns. It also accords with the central premise of our article, i.e., that aspects of WM map to different densely overlapping patterns of network activity/connectivity as opposed to mutually exclusive systems the brain. We hope you agree that these more temporally detailed analyses strengthen our article.

2) *The authors acknowledge the importance of, and have made some steps toward, integrating their results with the “classical,” “localist” literature, and this is appreciated, but there’s still a part of me that’s not sure that I understand the neural bases of working memory any better now than I did before reading this paper. Admittedly, part of this is due to the complexity and sophistication of the analytic methods, but for a general-readership journal like Nature Communications, one expects that the main point of the paper will be accessible to neuroscientists coming from a broad range of fields, and I just don’t think that this manuscript achieves this. Much of the writing comes across as so much “word salad,” epitomized by the final sentence of the manuscript.*

Our analyses are sophisticated, but we feel this is necessary and should be seen as a strength not a weakness. As recognised by Reviewer 2 *‘while some readers may find it difficult to follow the detailed aspects of the results, most readers should be able to comprehend the network-coding mechanism(s) being demonstrated’.*

Nonetheless, we agree with Reviewer 1 that many sections of the article communicated the results in a sub-optimal manner. The senior author has not extensively rewritten the article so that it should be accessible for a non-machine learning audience. Anything even vaguely resembling a word salad has been eliminated. The discussion section has been completely rewritten to explain clearly the most important implications of the findings.

3) *...there seem to be outright contradictions. For example, there’s the conclusion from Figure 3d that “domain-specific patterns of connectivity were enhanced as a function of WM load,” which is difficult to reconcile with the conclusions arising from Fig. 5 bi and bii. Looking at these figures the point seems to be that there are distinct connectivity patterns for Spatial vs. Fractal vs. Digit, but then a more-or-less common pattern of upregulation of connectivity between MD regions and visual regions is observed when load is increased, regardless of domain? It’s asserted that the latter pattern argues against ‘mutual exclusivity’ of discrete representation of domains. This would seem, on the face of it, to argue against the claims arising from Figure 3. And while I’m thinking of Fig. 5 bi and bii, isn’t an equally plausible description of these findings that domain stimulus domain is represented via very distinct “multiplex topological shapes [of connectivity patterns]” and then that, orthogonal to this, a different state that is common across domains waxes and wanes as the task gets more or less difficult?*

I think here the reviewer is referring to figure 4 as opposed to 3 as there is no figure 3d. The confusion here relates to the lack of a plain-speaking explanation of what was in the figures (apologies). The text and figures have been extensively revised (in places completely replaced) to address this issue. There actually are no contradictions in the results. The mass univariate analyses do indeed show common up regulation of connectivity at higher loads, but this is specific to the encode period of time. More stimuli are on the screen at this stage, so it is

unsurprising that connectivity within the visual system is generally higher.

There important analysis focuses on maintenance, when the visual display is the same for all loads. Here, the mass univariate analysis shows that there are no load main effects, only load*domain interactions and domain main effects. This suggests that the effects of *maintenance* load on connectivity is dependent on the stimulus domain that the maintained information is from. In further support of this view, the multivariate classification analysis operates more accurately at higher load; i.e., the multivariate patterns of connectivity become easier to dissociate.

It also is important to note that we do not find connections or nodes that are exclusively sensitive to one or other domain. E.g., in multiple demand cortex, they are generally more sensitive to different domains but active for all of them during maintenance. This accords with multivariate coding mechanism within this 'domain general' system, which is the main point that our article is making.

We understand that these are complicated concepts requiring sophisticated analyses and can see how our previous report of them has confused the reviewer. Therefore, we have extensively reworked the article in order to spell out in simple terms what each analysis means for the audience.

4) The Discussion asserts that one of the results “explains some of the confusion within the classic literature,” but then it’s not clear what this purported confusion is, nor are any citations offered.

We intended here to communicate (1) the confusion regarding how the wealth of information from the intensive early work that sought to map cognitive functions to modules in the brain may be reconciled with the modern network framework. Also (2) the debate regarding whether there are division of domain vs. divisions of process within the frontal lobes. There is now a discussion section that is dedicated to explaining how these contradictory theories may be reconciled when recast from a network-coding perspective.

5) there are important descriptive statistics missing: what were accuracies and RTs broken out by condition?

Tables detailing these results are included at the beginning of the supplementary materials. These are now called out from the main text.

6) Introduction: In the paragraph beginning “Second” I believe that the citations are flipped?

This section has been rewritten

Response to Reviewer 2

Reviewer #2 (Remarks to the Author):

Overall, the revised manuscript represents a significant improvement over the original version. The logic of the study is more clearly delineated, terms are more consistently presented, and the introduction/discussion sections are synthesized in a more easily comprehended manner. Furthermore, the authors did a lot of work to address specific concerns raised by all reviewers, including the new schematic (Fig. 6) that summarizes important results and distills the findings into a more cohesive narrative. That being said, select analyses, results, and visualizations are still fairly dense and difficult to parse out at times. However, the take away messages of the study are clearer now, and while some readers may find it difficult to follow the detailed aspects of the results, most readers should be able to comprehend the network-coding mechanism(s) being demonstrated. Moreover, while there are some concerns about “accessibility” of the findings, the questions addressed in this study along with the rigor of analyses are important for the field. Listed below are major and minor concerns moving forward; major concerns refer to resolving the “denseness” of the results section, and minor concerns are more technical in nature (i.e., more easily addressed).

Thank you for these encouraging comments and helpful suggestions - we agree that aspects of the results were not explained clear enough and that aspects of the visualisation were hard to follow. We have reworked the article extensively to make it as clear and accessible as possible.

Major concerns

1. Page 6: what is the logic behind switching focus to frontoparietal and visual ROIs? This might be clear to some readers, but 1 sentence explaining this would be helpful.

I think we are referring here to the mass univariate analysis - which was added in the second version of the article. The purpose of this was to determine (a) whether there are any connections that are sensitive to domain or load in general, or to domain*load interactions, especially during maintenance (where visual conditions are controlled). (b) having ascertained that there are no load-general connections, whether the effects of domain*load relate to connections within functionally distinct regions of the brain. In particular, a ‘top-down’ perspective would predict that maintenance involves frontoparietal areas that constitute SG to a greater extent than visual/motor areas. This is indeed the case. We have reworked the text of this section extensively.

2. Pages 6 – 7 and Figures 3 & 4: the details of these results were difficult to follow; more text on the maintenance-related findings would be helpful. For example, there was lowered global activity during maintenance relative to other stages, but connectivity analyses show more stability. However, there were influences/interactions based on which regions were in focus (visual and/or frontoparietal), cognitive load, inter/intra connectivity, and exactly which stages

were being compared. Overall, the complexity of this was hard to follow such that there was not a clear take-away from the results/figures. This was compounded by the classification analyses, which additionally raised the issue of how domain-general and stage-general patterns related to each other. A simple way to remedy this might be to connect the main-text more explicitly with the figures. This would ultimately add more text to the main body of the results section, but would help comprehension. For example, the distributions in Figure 3b-c could be explicated in the text as well (this suggestion could be applied throughout these sections – focusing on figures 3 & 4).

We agree - in retrospect the structure used in the ordering and legends for the figures were sub-optimal. We have extensively reworked this throughout the article and linked them together more explicitly. We also have rewritten all of the figure legends so as to be more accessible for a general readership.

Minor concerns

1. *General: consider having someone triple check that the panel letters for figures match what is in the text and that all panel letters are consistently written in figure captions*

We have done this

2. *General: probe/recall/retrieve seem to be used as synonyms; consider choosing 1 term for consistency*

We have changed this to 'probe' throughout

3. *Page 2, line 20: the term phonological is used here but nowhere else; consider replacing it with 'number' to be more consistent with what the study is directly manipulating*

We have replaced with 'number' throughout

4. *Page 4, line 9: "fractal patterns" should be singular for consistency*

We have replaced with the term 'fractal' or 'fractal(s)' throughout

5. *Page 5, line 19: "study 2" in this syntax seems to be missing*

We have fixed this

6. *Page 5, lines 14-21: consider using the past tense exclusively*

We have endeavoured to switch text to past tense throughout results, methods and figure legends.

7. *Page 17, line 24: capitalize "procedure"*

This has been done

8. Methods section general: use consistent tense

We have switched to consistent past tense

9. Page 28, Figure 3a: consider adding a scale for the connectivity matrices

This has been added

10. Page 28, Figure 3c: consider stating which regions were used in the averages here

This is now explained in the figure legend

11. Page 29, Figure 4: the text is cut off in the pdf

This has been fixed

12. Page 29, Figure 4aii-iii: consider writing what colors indicate

We have updated this figure and replaced the legend

13. Page 31, Figure 6: typo in 'parcellation sets' panel; fix to "unbiased"

This figure has been reworked

14. Page 31, Figure 6: random words seem to be capitalized in the text descriptions

Text also has been updated for this figure.

We apologise for this abundance of typos - these and others have been fixed in the substantially reworked manuscript.

REVIEWERS' COMMENTS:

Reviewer #1 (Remarks to the Author):

The authors have satisfactorily address the points that I raised in the previous round of reviews.